# A giant amphipathic helix from a perilipin that is adapted for coating lipid droplets

Alenka Čopič [1], Sandra Antoine-Bally[1], Manuel Giménez-Andrés [1,2], César La Torre Garay[1], Bruno Antonny [3], Marco M. Manni [3], Sophie Pagnotta[3], Jeanne Guihot[1] & Catherine L. Jackson [1]

How proteins are targeted to lipid droplets (LDs) and distinguish the LD surface from the surfaces of other organelles is poorly understood, but many contain predicted amphipathic helices (AHs) that are involved in targeting. We have focused on human perilipin 4 (Plin4), which contains an AH that is exceptional in terms of length and repetitiveness. Using model cellular systems, we show that AH length, hydrophobicity, and charge are important for AH targeting to LDs and that these properties can compensate for one another, albeit at a loss of targeting specificity. Using synthetic lipids, we show that purified Plin4 AH binds poorly to lipid bilayers but strongly interacts with pure triglycerides, acting as a coat and forming small oil droplets. Because Plin4 overexpression alleviates LD instability under conditions where their coverage by phospholipids is limiting, we propose that the Plin4 AH replaces the LD lipid monolayer, for example during LD growth.

[1] Institut Jacques Monod, CNRS, UMR 7592, Université Paris Diderot, Sorbonne Paris Cité, 75013 Paris, France. [2] Université Paris-Sud, Université Paris-Saclay, 91405 Orsay, France. [3] Université Côte d'Azur, CNRS, IPMC, 06560 Valbonne, France. These authors contributed equally: Sandra Antoine-Bally, Manuel Giménez-Andrés, César La Torre Garay. Correspondence and requests for materials should be addressed to A.Č. (email: alenka.copic@ijm.fr)

Lipid droplets are ubiquitous cellular organelles that serve as the primary depot for energy and lipid storage in eukaryotic cells. As such, they play an important role in the maintenance of cellular homeostasis and their malfunction is associated with numerous diseases, from obesity and diabetes to cancer and neurodegenerative diseases[1–3]. How proteins that mediate LD function (for example enzymes and regulators of lipid metabolism) are selectively targeted to the surface of this organelle is poorly understood[4].

LDs are composed of a neutral lipid core, consisting primarily of triglycerides and sterol esters, which is covered by a monolayer of phospholipids and other amphiphilic lipids, and by proteins[4,5]. Unlike the two leaflets of a bilayer, which are physically coupled, this monolayer can be stretched infinitely, resulting in an increasing surface tension. LDs with high surface tension are unstable and tend to fuse; LD fusion appears to be one mechanism by which cells can cope with an imbalance in synthesis of neutral lipids and phospholipids[6–8]. LD size appears regulated and is highly variable between different cells, ranging from 100 nm to 100 μm, with mature adipocytes often containing only one large LD[4,9]. On the other side of the size spectrum are the small lipoprotein particles, which, like LDs, are adapted for harboring neutral lipids, but are secreted from cells or form extracellularly[10].

Numerous LD proteins contain regions that are predicted to form amphipathic helices (AHs), and it has been shown in many cases that these regions are important for LD targeting[6,11–13]. AHs are also involved in targeting of proteins to other cellular organelles. They have been shown to specifically recognize different features of lipid bilayers, such as surface charge, packing of acyl chains, and membrane curvature[14–17]. Lipid packing and membrane curvature promote AH recruitment through creation of lipid packing defects, which have been analyzed extensively in bilayer membranes[18–20]. It is not clear which parameters are important for AH binding to the LD surface[21]. A recent in silico analysis suggests that whereas an LD phospholipid monolayer at zero surface tension does not behave very differently from a bilayer, lipid packing defects increase non-linearly with increasing surface tension of the monolayer[22]. This result could explain why under some experimental conditions, AHs have been observed to bind to LDs rather non-discriminately[23–25]. Finally, recruitment of proteins to the LD surface also appears sensitive to protein crowding[26,27].

We aim to understand to what extent AHs can be selective for the LD surface and what parameters are important for this selectivity. The AHs that have been shown to localize to LDs appear highly diverse, precluding any speculation about AH localization based on sequence comparisons[6,11,13,23,25,28]. We have instead focused on one particular AH present in the mammalian LD protein perilipin 4 (Plin4). Plin4 is related in its sequence to the other mammalian perilipins (Plin1–5), which all localize to LDs and interact with lipid enzymes and other regulators of LD metabolism[29]. The carboxy-terminal portions of perilipins are predicted to fold into a four-helix bundle, which has been crystalized in Plin3[30], whereas in their amino terminal/central region they all contain an 11-mer repeat sequence[31]. These 11-mer repeat regions have been shown to mediate LD localization of Plin1–3 and can form an AH[11,13]. Interestingly, these features are also present in apolipoproteins, which form small lipoprotein particles, but it is not clear whether the two protein families are evolutionarily related[30–32]. Plin4 is the least explored of all perilipins; it is highly expressed in adipocytes, where it may associate preferentially with small LDs, but it is absent from most other tissues and its physiological role is not clear[33–36]. However, the 11-mer repeat sequence of Plin4 is exceptional in terms of its length and repetitiveness (Fig. 1).

We now show that the Plin4 11-mer repeat region localizes to LDs in different cellular models and can directly interact with neutral lipids in vitro. The properties of this giant AH have allowed us to manipulate it in a modular manner to dissect the parameters that control its localization in cells. We show that the length, hydrophobicity, and charge of the AH all contribute to its LD localization and that these properties can to some extent compensate for one another. Finally, we show that overexpression of the Plin4 AH can rescue an LD size defect associated with cellular depletion of phosphatidylcholine (PC)[6,7], suggesting that the ability of this AH to interact with neutral lipids may be important for its in vivo function.

## Results

**Plin4 contains a giant AH that localizes to LDs.** The length of the predicted AH in Plin4 surpasses that of other known AHs involved in organelle targeting by an order of magnitude (Fig. 1a and Table 1). The defining feature of this region is the presence of 11-mer repeats; if folded, these repeats could adopt a slightly extended version of an α-helix termed a 3–11 helix (Fig. 1b). This type of helix has been well characterized in apolipoproteins and in α-synuclein, where the folding is dependent on contact with lipids[31,37,38]. Furthermore, the repeats in Plin4 are extremely well conserved at the level of 33-mer (3×11-mer): 29 tandem 33-mer repeats can be identified in the human Plin4 sequence, entirely without deletions or insertions between them (Fig. 1c, d). These striking properties make Plin4 an ideal model to study the parameters that govern AH targeting to LDs.

In contrast to its exceptional length and monotonous composition, the Plin4 AH is weak in its amphipathic character when compared to other well-studied AHs (Table 1 and Supplementary Fig. 1). For our analysis of Plin4 AH targeting in cells, we chose the most conserved central portion of the predicted AH sequence, where the small differences between the 33-mer repeats can be considered negligible (Fig. 1 and Supplementary Fig. 1b). We expressed different fragments of the protein as fluorescent protein fusions in HeLa cells, which do not express endogenous Plin4 (Fig. 1d, e)[39]. Whereas a peptide comprising 66 amino acids of the Plin4 AH, i.e., two 33-mers, was completely cytosolic, increasing the length to 4×33-mer repeats (132 aa) or 8×33-mers resulted in a fraction of LDs that were positive for Plin4. Further extending the length of the AH to 20×33-mer repeats (660 aa) led to its localization to all LDs (Fig. 1e, f). The size and number of LDs in cells was not significantly affected by the expression of these constructs (Supplementary Fig. 2a). In all cases, an appreciable amount of cytosolic signal could also be observed, independent of the amount of LDs in cells (cells were either grown in standard medium or supplemented with oleic acid to induce LD accumulation, Supplementary Fig. 2b). In contrast to the AH region, the C-terminus of Plin4 containing the predicted 4-helix bundle was completely cytosolic (Supplementary Fig. 2c). The experiments presented so far were performed on fixed cells, and we found that fixation itself augmented Plin4 AH targeting to LDs, presumably by stabilizing the protein on LDs. However, a similar trend of improved LD targeting with increasing length of constructs could be observed in live cells (Supplementary Fig. 2d).

We confirmed the LD surface targeting of the Plin4 AH by expressing it in budding yeast, which contain proteins that are distantly related to human perilipins[27,40]. Similar to HeLa cells, a robust LD signal could be observed with Plin4-12mer, whereas Plin4-4mer remained largely cytosolic (Fig. 1g). The fact that the same sequence is targeted to LDs in such evolutionarily distant organisms speaks to the universal nature of AH–LD interactions, as previously shown[13,40]. In addition, the Plin4-12mer was also

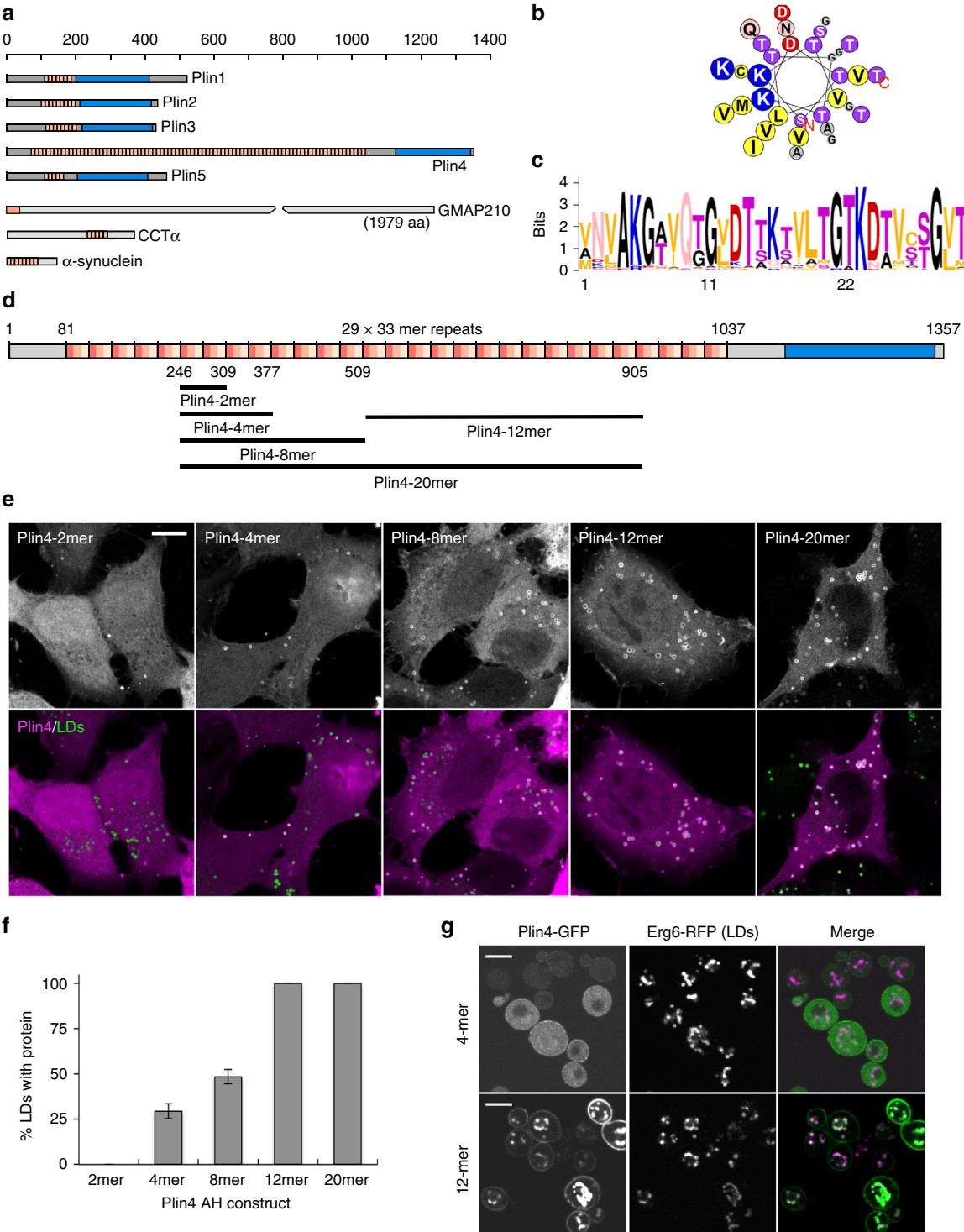

**Fig. 1** Plin4 contains a very long AH that localizes to LDs. **a** Schematic diagrams of human proteins with long AH regions (pink). In the proteins of the perilipin family (Plin1–5), CTP:Phosphocholine Cytidylyltransferase α (CCTα) and α-synuclein, in which the AH region contains 11-mer repeats, the corresponding bars are segmented to indicate the approximate repeat number. In Plin4, these repeats are remarkably conserved at the level of 33-mers. All perilipins also contain a predicted 4-helix bundle (blue), which has been crystalized in Plin3. **b** Helical wheel plot of one 33-mer repeat from Plin4, plotted as a 3–11 helix[71]. **c** Weblogo generated from an alignment of 29 33-mer repeats from human Plin4 sequence[72]. **d** Schematic representation of human Plin4. Position and length of different Plin4 constructs is also shown. **e** Localization of Plin4-AH-mCherry constructs of different length in HeLa cells (upper panels). Lower panels show colocalization of Plin4-AH (magenta) and LDs stained with Bodipy (green). Scale bar: 10 μm. **f** Quantification of the mean fraction of LDs stained with Plin4-AH per cell in HeLa cells. 40–60 cells in two independent experiments were quantified for each construct. Error bars depict s.e.m. from one experiment (the range between the two experiments is smaller than this error bar). **g** Localization of Plin4-4mer and Plin4-12mer GFP fusions in yeast. Lipid droplets are marked with Erg6-RFP. Scale bar: 5 μm

**Table 1 Comparison of properties of different AHs**

| AH | Position[a] | Length[b] | Hydr. mom[c] | Fract of residues[d] | | | Lipid-binding properties | Ref |
|---|---|---|---|---|---|---|---|---|
| | | | | Hdr | LH | Ch | | |
| Hel 13-5 | n.a.[e] | 18 | 0.69 | 0.72 | 0.06 | 0.28 | Tubulates liposomes | 73 |
| GMAP210 | 1–38 (1979) | 38 | 0.48 | 0.37 | 0.05 | 0.05 | Golgi vesicles/abundant packing defects | 14, 25 |
| Nup133 | 247–267 (1156) | 21 | 0.44 | 0.38 | 0.05 | 0.10 | Nuclear pore membrane | 67 |
| CIDEA | 163–180 (219) | 18 | 0.53 | 0.5 | 0.11 | 0.17 | LDs[f] | 28 |
| CCTα | 236–294 (368) | 59 | 0.48* | 0.32 | 0.10 | 0.49 | Nuclear envelope/ER and LDs/liposomes | 6, 16, 64 |
| Apolipoprotein-AI | 74–267 (267) | 186 (7 P)[g] | 0.39* | 0.40 | 0.065 | 0.35 | Lipoprotein particles | 48 |
| α-Synuclein | 1–89 (144) | 89 | 0.30* | 0.47 | 0.02 | 0.24 | Synaptic vesicles/neg. charged small liposomes | 44 |
| Plin3 | 114–204 (434) | 90 | 0.35* | 0.37 | 0.01 | 0.23 | LDs and cytosol | 11 |
| Plin4 | 70–1037 (1357) | 968 | (0.257)* | 0.35 | 0.002 | 0.15 | This study | |

[a] Numbers give the first and last amino acid in the protein sequence, with the total length of each protein given in brackets
[b] Length is based on structural data, when available, and visual inspection using Heliquest[71]. Prolines are considered as helix breakers
[c] Hydrophobic moment is calculated using Heliquest[74]. The AHs that contain 11-mer repeats are plotted as 3–11 helices. For longer AHs (*), the mean of hydrophobic moments of consecutive helices is given (see Supplementary Fig. 1 for details)
[d] Fraction of different amino acids in the AH sequence: Hdr, all hydrophobic (A, I, L, M, V, F, W, Y); LH, large hydrophobic (aromatic) (F, W, Y); Ch, charged=acidic (D, E) and basic (K, R)
[e] This is an artificial AH
[f] This AH has only been tested as a fusion with other parts of the protein, therefore its specificity for LDs is not known
[g] The helix is broken by seven prolines, which induce a kink; ten separate helices in the structure (Supplementary Fig. 1)

detected at the periphery of yeast cells. This is in accordance with the net positive charge of this amphipathic sequence (Fig. 1b), which could be interacting electrostatically with the negatively charged plasma membrane[41,42]. A fluorescent signal could also sometimes be observed at the periphery of HeLa cells, but due to cell shape and a large pool of cytoplasmic protein, this signal was difficult to quantify. Endogenous Plin4 has been detected at the cell periphery in adipocytes, in addition to its localization to LDs[33,35,43]. Furthermore, the Plin4 AH is related in its chemistry to the AH of α-synuclein, a protein enriched in presynaptic termini and on negatively charged vesicles[44] (Supplementary Fig. 1). Interestingly, α-synuclein has also been observed on LDs when expressed heterologously[44,45], and the closely-related γ-synuclein localizes to LDs in adipocytes, where it affects lipid metabolism[46]. An α-synuclein AH construct containing 85 amino acids localizes to LDs slightly better than Plin-4mer (Supplementary Fig. 2e, f), consistent with the higher hydrophobic moment of the α-synuclein AH (Table 1).

Overall, our analysis indicates a correlation between the targeting of the Plin4 AH to LDs and its length: longer constructs interact better owing to a more extensive interaction surface. A relatively long Plin4 sequence (>66 aa) is required for targeting to LDs, indicating that the elementary 33-mer repeat of Plin4 is a selective but intrinsically weak determinant for LD targeting. In this respect, we note that the Plin4 33-mer repeat displays the smallest hydrophobicity and hydrophobic moment among all AHs known to interact with lipid surfaces (Table 1)[47].

**The Plin4 AH is intrinsically unfolded in solution**. Membrane-interacting AHs are often unfolded in solution, adopting a helical structure only upon contact with membranes, as exemplified by the AH of α-synuclein[38]. Alternatively, the AH may be folded into a different structure in the soluble form of its parent protein, as is for example the case for CTP:Phosphocholine Cytidylyl-transferase α (CCTα) and Arf1[16,37,48,49]. To study the biochemical properties of the different Plin4 AH fragments, we expressed and purified them from *Escherichia coli*, following the procedure that was used for purification of α-synuclein[50]. The Plin4 4mer, 12mer, and 20mer in bacterial lysates were resistant to boiling, and they could subsequently be purified from the remaining contaminants by anion-exchange chromatography

(Supplementary Fig. 3a, b). All constructs eluted from the column at the same salt concentration, as expected considering their similar sequences.

We used size exclusion chromatography to characterize the hydrodynamic properties of the three Plin4 constructs. They migrated on the column at the same elution volume as well-folded protein standards with a twofold higher MW (Fig. 2a, b). Because such an increase in apparent MW (i.e., Stokes radius) is observed for many proteins upon denaturation[51], this result suggests that all Plin4 constructs are intrinsically unfolded. The lack of secondary structure was confirmed by circular dichroism (CD) spectroscopy (Fig. 2c). However, Plin4 fragments adopted a highly helical conformation when incubated in 50% trifluoroethanol solution. These experiments indicate that Plin4 contains a giant repetitive region that is intrinsically unfolded but has the potential to fold into a helix much longer than any other previously described AH (Table 1).

**Plin4 AH targeting to LDs is controlled by hydrophobicity**. The amino acid composition of the Plin4 AH region is very particular: the segment containing close to 1000 aa is almost devoid of aromatic residues (accounting for only 0.6% of all residues in the human Plin4 AH sequence, about 20 times less than is the vertebrate average), whereas some amino acids, in particular threonine, glycine, and valine, are highly enriched (Supplementary Table 1)[52]. These features contribute to the low hydrophobicity and low-hydrophobic moment of the predicted AH (Table 1).

Given the importance of Plin4 AH length for LD targeting, we surmise that its binding to LDs is mediated by multiple weak interactions over an extended binding surface. To probe the nature of these interactions, we devised a mutagenesis strategy whereby subtle mutations were repeated along the helix. As the starting point, we used the Plin4-4mer, whose length is just above the threshold for LD binding and thus the targeting of this helix may be sensitive to even small perturbations. All mutants were expressed in HeLa cells as mCherry fusions in the same manner as the wild-type form. The LD targeting phenotypes of the different constructs did not depend on their expression levels.

We mutated threonine residues in the non-polar face of the helix into the structurally similar but more hydrophobic valine (Fig. 3a and Supplementary Table 2). A single threonine to valine

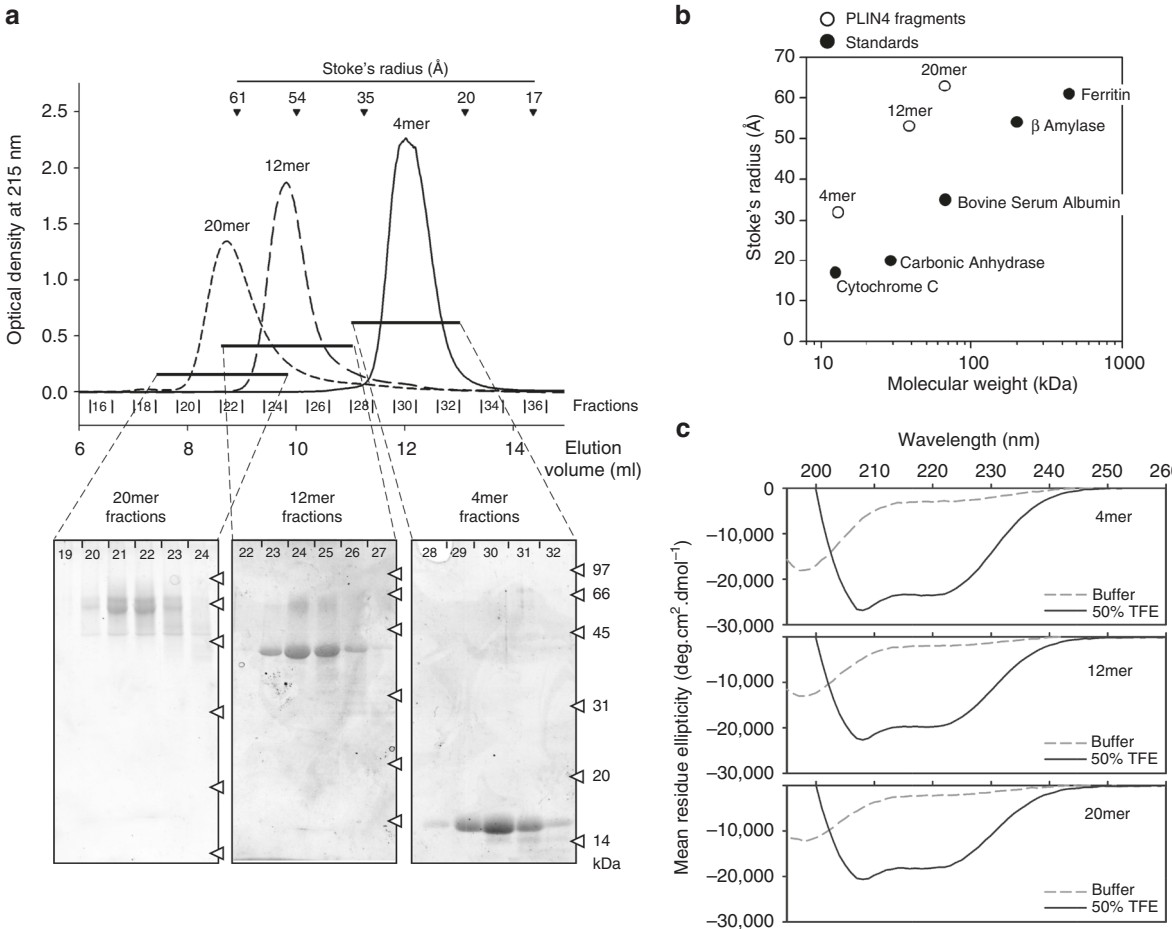

**Fig. 2** Hydrodynamic and secondary structure analysis of Plin4 AH fragments. **a** Elution profiles of the 4mer, 12mer, and 20mer fragments of Plin4 on a Superose 12 column. The black arrowheads indicate the elution volumes of protein standards of defined Stoke's radii, which were used to calibrate the column. The indicated fractions were analyzed by SDS-PAGE with Sypro Orange staining. The white arrowheads indicate the migration of molecular weight (MW) standards on the gels. **b** Plot of the apparent Stoke's radius vs MW for Plin4-4mer, 12mer, and 20mer (white circles) and protein standards (black circles) as determined from the chromatograms shown in **a**. The Stoke's radius of the Plin4 fragments is about twofold higher than that of folded proteins of similar MW. **c** CD analysis of the Plin4 fragments. The spectra were acquired either in buffer (dashed gray lines) or in an equal volume of buffer and trifluoroethanol (black lines). PLIN4 concentration: 4mer, 19 μM; 8mer, 7.5 μM; 20mer, 4 μM

substitution in each of the 33-mers (1T→V mutant) resulted in a marked improvement in LD localization, as now all LDs in transfected cells became positive for mutated Plin4 AH. The same was true for 2T→V and 3T→V mutations, and even for the only slightly more hydrophobic 3T→A construct (Fig. 3b, c). The T→V mutants could also be observed on LDs in live cells (Supplementary Fig. 4a). In contrast, a less hydrophobic 4T→S mutant with the same number of hydroxylated residues was completely cytosolic under all experimental conditions (Fig. 3b, c).

Concomitant with an increase in their affinity for LDs, the hydrophobic mutants became more promiscuous for binding to other cellular membranes. First, unlike the wild-type Plin4-4mer, more hydrophobic mutants did not stain the nucleus, suggesting that the non-LD localized Plin4 pool has shifted from soluble to membrane-bound (Fig. 3b, d). Note that the Plin4-4mer fused to mCherry is small enough to freely diffuse from the cytosol to the nucleus. Second, a strong reticular signal was observed in cells that highly expressed the 3T→V mutant, whereas in low-expressing cells the signal was much brighter on LDs than in the cytoplasm, suggesting that this AH saturates the LDs before invading other membranes (Fig. 3e). Note that the degree of AH localization to LDs was independent of protein expression levels. Finally, the 3T→V mutant colocalized well with the endoplasmic

reticulum (ER) marker Sec61b, in contrast to the wild-type Plin4-4mer (Supplementary Fig. 4b). We conclude that the low hydrophobicity of the Plin4 AH permits a reversible interaction with LDs over a long interaction surface while minimizing the AH binding to other cellular membranes.

**Charge of the Plin4 AH affects its targeting**. We next analyzed the polar face of the predicted Plin4 AH, which is equally striking in composition. Almost systematically, each 33-mer repeat contains three positively and two negatively charged amino acids. We reversed the net charge of the AH from +1 to −1 by mutating 2 lysines per 33-mer to glutamine, again aiming to minimize other changes to helical properties (Fig. 4a and Supplementary Table 2). Despite being more hydrophobic, the 2K→Q–mCherry fusion became cytosolic and absent from all LDs (Fig. 4b, c). However, its targeting to LDs could be rescued by a further increase in hydrophobicity via the addition of the previously tested 2T→V substitution, showing that the helix is still functional. These results suggest that, whereas net positive charge of the AH is advantageous for LD targeting, it is not essential: a negatively charged AH can also localize to the LD surface, excluding a large contribution from electrostatic interactions and consistent with

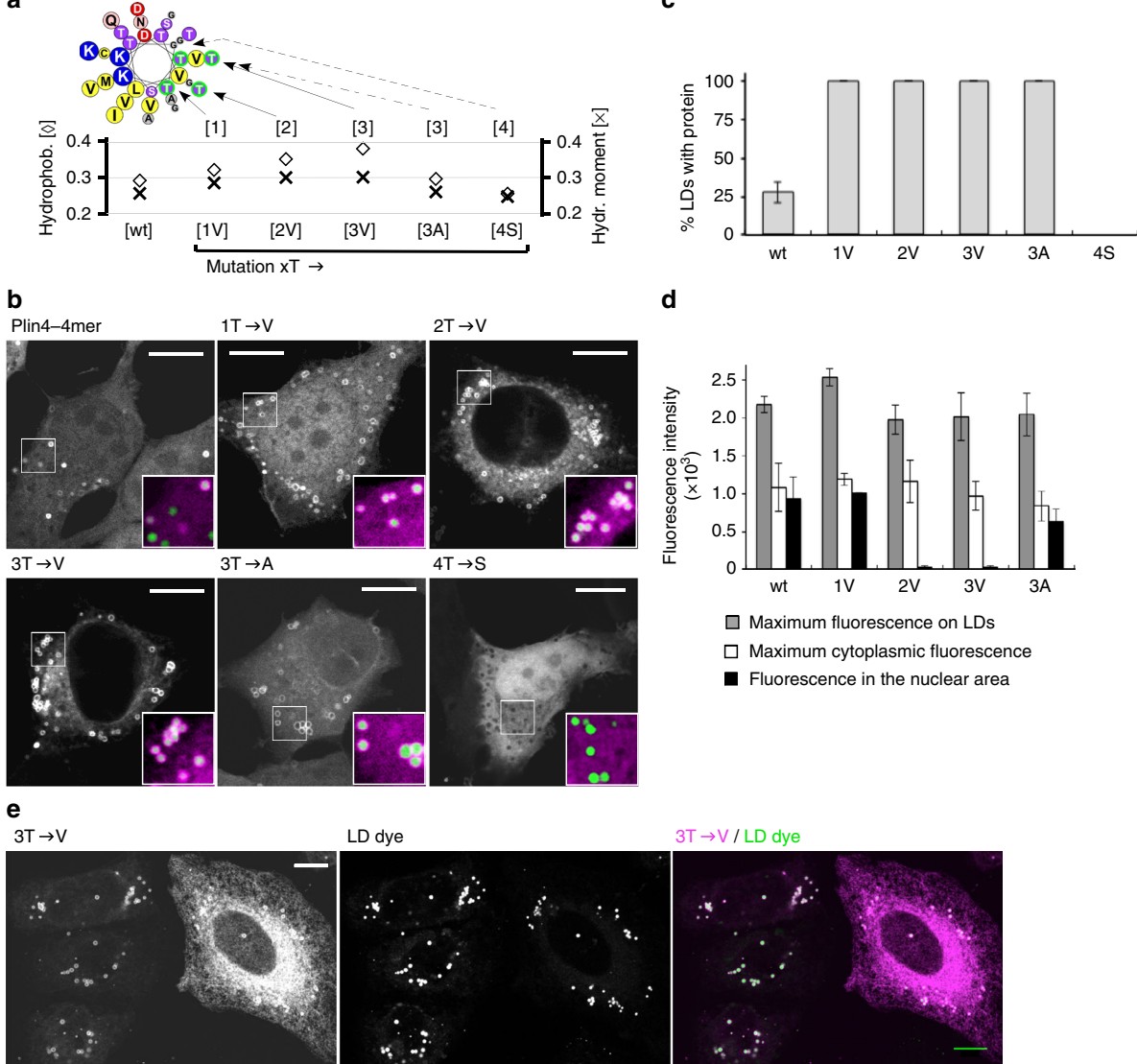

**Fig. 3** Influence of AH hydrophobicity on the selective targeting of Plin4 to LDs. **a** Helical wheel depicting cumulative mutations in the hydrophobic face of a Plin4 33-mer. The plot shows the hydrophobicity and hydrophobic moment of each mutant as calculated using Heliquest[71]. **b–d** Analysis of the mutants in the context of the Plin4-4mer in HeLa cells described in **a**. For example, the 2T→V mutant corresponds to a construct containing four identical 33mer repeats with two threonine to valine substitutions per repeat. All constructs were expressed as mCherry fusions. **b** Representative examples with large image showing mCherry staining and insets showing merge between the mCherry signal and the LD dye Bodipy. **c** Fraction of LDs per cell positive for the indicated Plin4 construct. **d** Quantification of maximum fluorescence on LDs, maximum fluorescence in the cytoplasm (excluding LDs), and mean fluorescence in the nuclear area for each Plin4 construct. 30–40 cells per experiment were quantified for each construct, and the error bars depict the range of means between two independent experiments. **e** At higher expression level, the 2T→V (not shown) and 3T→V mutants strongly stain the ER network in addition to LDs. In contrast, LD staining is similar between cells with different level of Plin4 expression. Scale bar: 10 μm

lipidomic analysis of LDs[53]. When we expressed the 2K→Q, 2T→V mutant in budding yeast, it could likewise be observed on LDs (Fig. 4d). In contrast, the 2K→Q mutation largely prevented localization of the AH to the negatively charged plasma membrane, where electrostatic interactions are important for protein targeting.

We then constructed an inverse type of mutant, 2D→N, in which we mutated all acidic residues to glutamine (Fig. 4a). As this mutant is more positively charged, it should bind better to negatively charged surfaces[14–16]. Strikingly, we observed a decrease in LD localization, which could be rescued by the 2T→V substitutions (Fig. 4b, c), suggesting that acidic residues also contribute to LD-Plin4 association.

If electrostatic interactions between the Plin4 AH and the lipid surface of LDs are not essential, why are charged residues so conserved throughout the AH sequence? We have noted that charge is always distributed asymmetrically in the polar face of the Plin4 helix, with lysines clustering on one side. This organization is unusual and is not optimal for interaction between lysines and a lipid surface[14,54]. We reorganized the charged residues in the Plin4 AH to make them more symmetrically distributed while maintaining the same amino acid composition and minimizing the change in hydrophobic moment (Fig. 4b). The resulting 'charge-swap' mutant could not be detected on LDs when expressed in HeLa cells. As with the previous charge mutants, the LD-targeting of the charge-swap AH could be rescued by the 2T→V substitution (Fig. 4d). Based

on these observations, we hypothesize that charged residues in Plin4 could be mediating inter-helical interactions to stabilize the AHs on the LD surface, making recruitment of this AH cooperative. Cooperative binding is consistent with our observation that even for helices that are recruited relatively poorly to

LDs, the maximum intensity of fluorescent signal on LDs is similar to that of the more hydrophobic AHs of the same length (Fig. 3b, d).

Overall, the mutational analysis of the polar face of the Plin4 AH indicates that the targeting of Plin4 to LDs strongly departs

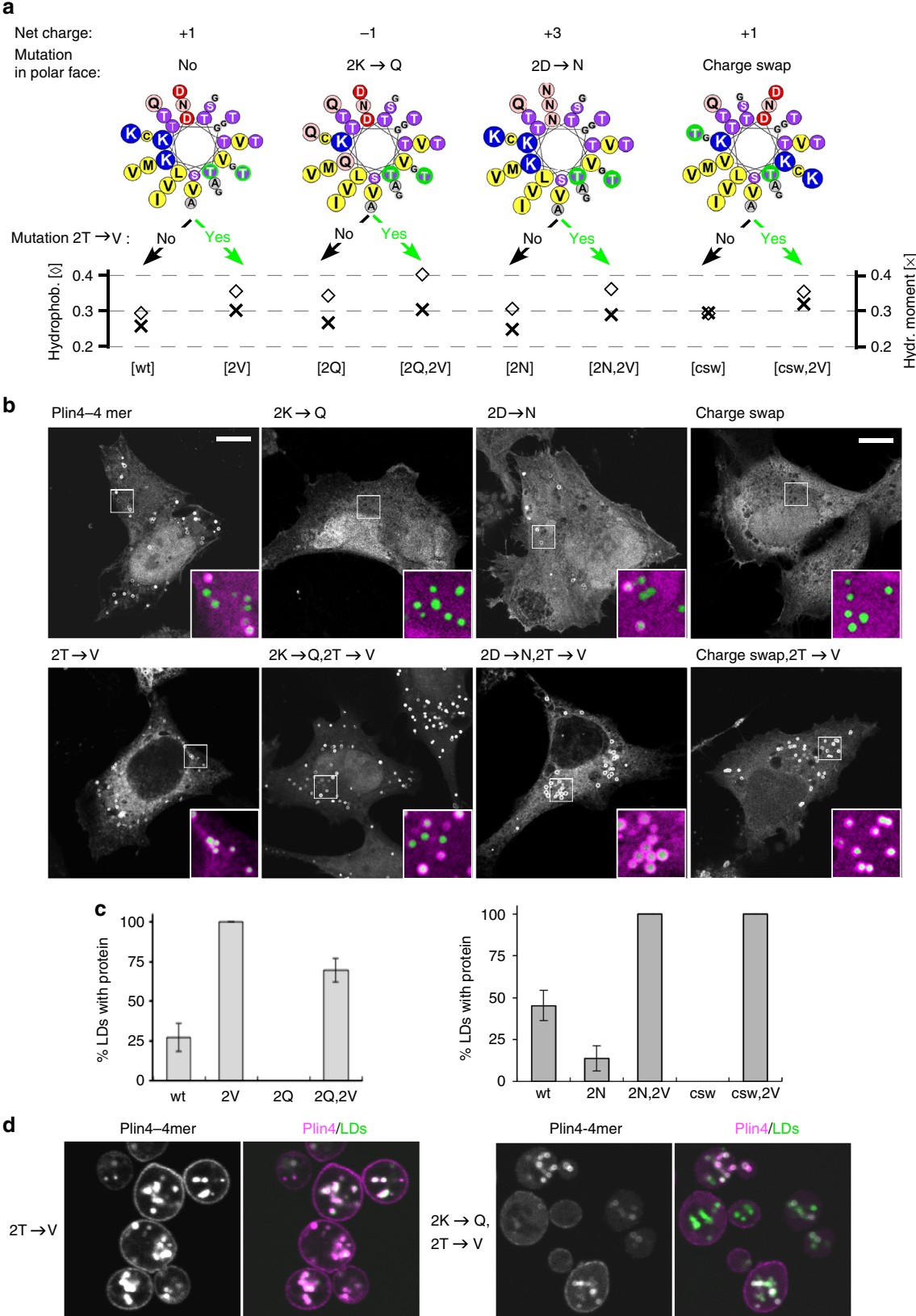

from the mechanism by which many AHs interact with bilayers. In general, positively charged residues electrostatically interact with negatively charged phospholipids. In Plin4, instead, both positively and negatively charged residues contribute to LD targeting and this contribution requires these residues to be properly distributed.

**Purified Plin4 AH interacts poorly with bilayer membranes.** We used purified Plin4 AH constructs to dissect how they interact with artificial lipid surfaces. Proteins were labeled with NBD or Alexa fluorescent probes via cysteine residues that are present in the Plin4 AH sequence (Fig. 5a, Supplementary Fig. 3c). NBD fluorescence increases in a non-polar environment, allowing quantitative measurement of membrane association.

We performed binding experiments with liposomes of defined size and composition, and varied two parameters that influence the recruitment of many AHs to bilayers: membrane electrostatics and lipid packing (Fig. 5b–e)[14,17,18]. For electrostatics, we replaced PC with phosphatidylserine (PS). For lipid packing, we modulated the amount of lipid packing defects (i.e., exposure of non-polar hydrocarbons) in three different ways. First, we increased lipid unsaturation by replacing saturated-monounsaturated (C16:0-C18:1) phospholipids with their dioleyl equivalents (18:1-C18:1). Second, we increased membrane curvature by decreasing the pore size of the filters used for liposome extrusion. Third, we included the unusual methyl-branched diphytanoyl phospholipids. These lipids, which are not present in eukaryotic cells, have been recently shown to strongly promote α-synuclein adsorption to liposomes by inducing a more drastic increase in lipid packing defects than curvature or lipid mono-unsaturation[55]. In all cases, NBD-labeled proteins were incubated with liposomes at a low protein-to-lipid ratio to minimize crowding effects. For comparison, we used α-synuclein, whose association with bilayers has been studied extensively[17,38,44].

For most liposomes tested, the NBD fluorescence of wild-type Plin4-4mer remained close to that observed in solution suggesting very weak membrane adsorption. These included liposomes with a high content of dioleoyl lipids, negatively charged liposomes, and small liposomes (Fig. 5b–e, black symbols). Furthermore, Plin4 AH was not recruited to liposomes containing an increasing concentration of dioleoyl-glycerol (DOG) (Fig. 5f). DOG was shown to promote Plin3 localization to the ER or to ER-associated LDs[56,57]. The only exception was liposomes containing diphytanoyl phospholipids, for which we observed a dramatic increase in the fluorescence of NBD-Plin4-4mer (Fig. 5b). Flotation experiments indicated total binding of all NBD constructs to 100% diphytanoyl-PS liposomes, thus we used these liposomes to calibrate the fluorescence signal and determine the percentage of protein recruitment under all conditions. In contrast to wild-type Plin4-4mer, the 2T→V mutant displayed a marked increase in fluorescence in the presence of liposomes, and its fluorescence was further augmented by increasing charge or amount of membrane packing defects (Fig. 5b–e, green symbols). α-synuclein, behaved as was previously shown[17]: it interacted

with charged and highly curved liposomes, but displayed no binding to neutral liposomes regardless of their degree of unsaturation (Fig. 5b–e, blue symbols).

All Plin4-AH wild-type constructs interacted with diphytanoyl liposomes and their binding affinity increased with AH length, similar to the effect on LD targeting in cells (Figs. 5g and 1). Finally, we used diphytanoyl liposomes to test whether the Plin4 repetitive sequence can fold into a helix in contact with a lipid surface. The addition of such liposomes to the Plin4 peptide induced a large CD peak at 220 nm, indicative of very high α-helical content (≈70%, Fig. 5h). The result shown was achieved with the Plin4-20mer peptide that contains 660 amino acids, which is far longer than other α-helices with a similarly characteristic CD signature.

In conclusion, purified Plin4 AH interacts poorly with most bilayers, but small and repetitive modifications in its amino acid composition can dramatically improve membrane binding in a non-selective manner, as demonstrated by the slightly more hydrophobic 2T→V mutant. In contrast, α-synuclein displays selectivity for bilayers combining curvature and electrostatics (Fig. 5c–e)[17]. The amino acid composition of Plin4 AH therefore seems tuned to exclude most lipid bilayers, which fits well with the low staining of membrane-bound organelles in vivo. However, this lack of avidity for lipid bilayers does not explain why Plin4 selectively adsorbs to lipid droplets.

**Plin4 can bind directly to neutral lipids.** Given our observations that Plin4 AH selectively binds to LDs in cells and to diphytanoyl liposomes in vitro, we considered possible similarities between these structures. Due to the presence of methyl side chains, diphytanoyl phospholipids form a very poorly packed bilayer with a high degree of exposure of acyl chain carbons (i.e., lipid packing defects)[55]. An in silico analysis demonstrates a sharp increase in lipid packing defects on the phospholipid monolayer of a model LD under conditions of increased surface tension (i.e., an increase in neutral core to surface phospholipid ratio)[22]. Furthermore, increased surface tension promotes α-synuclein binding to inverted LDs in vitro[23]. At the extreme end, an LD completely devoid of a phospholipid monolayer (consisting of only neutral lipids) can be imagined as a lipid surface with infinite packing defects (acyl chain carbons exposed over the whole surface). We therefore asked if the Plin4-AH is capable of interacting with neutral lipids in the absence of any phospholipids.

We mixed a drop of a triglyceride, triolein, with purified Plin4-12mer at increasing protein concentrations, up to a protein-to-lipid molar ratio 1:2000, although there is some uncertainty in protein concentration (Methods). After vigorous vortexing, the suspensions became turbid, suggesting that the oil was emulsified into smaller droplets (Fig. 6a). Vortexing did not affect the properties of the Plin4 AH (Supplementary Fig. 5a, b). Electron microscopy of the Plin4-triolein emulsion revealed numerous spherical droplets with a diameter of 50 to 250 nm (Fig. 6b, Supplementary Fig. 5c). Dynamic light scattering measurements confirmed the presence of particles in the same size range (Fig. 6c,

**Fig. 4** Influence of AH charge on the selective targeting of Plin4 to LDs. **a** Helical wheels depicting the mutations that were introduced into Plin4 33mer. In this series, charged residues in the polar face were mutated, and the hydrophobic face was either kept intact or modified with the previously characterized 2T→V mutation (Fig. 3). Charge-swap mutant is abbreviated as 'csw'. The plot shows the hydrophobicity and hydrophobic moment of each mutant as calculated using Heliquest[71]. **b** Representative images of the mutants expressed as mCherry fusions in HeLa cells LDs were stained with Bodipy. All mutants were prepared as identical 4-mer repeats. Scale bar: 10 μm. **c** Quantification of HeLa images, showing % of LDs per cell positive for the indicated Plin4 construct. 30–40 cells per experiment were quantified for each construct and the error bars depict the range of means between two independent experiments. Two sets of mutants were analyzed at different times in the project and are therefore presented on separate plots. **d** Representative images of the mutants expressed as GFP fusions in budding yeast. LDs were marked with Erg6-RFP. For consistency, the colors of the yeast images are inverted. Scale bar: 5 μm

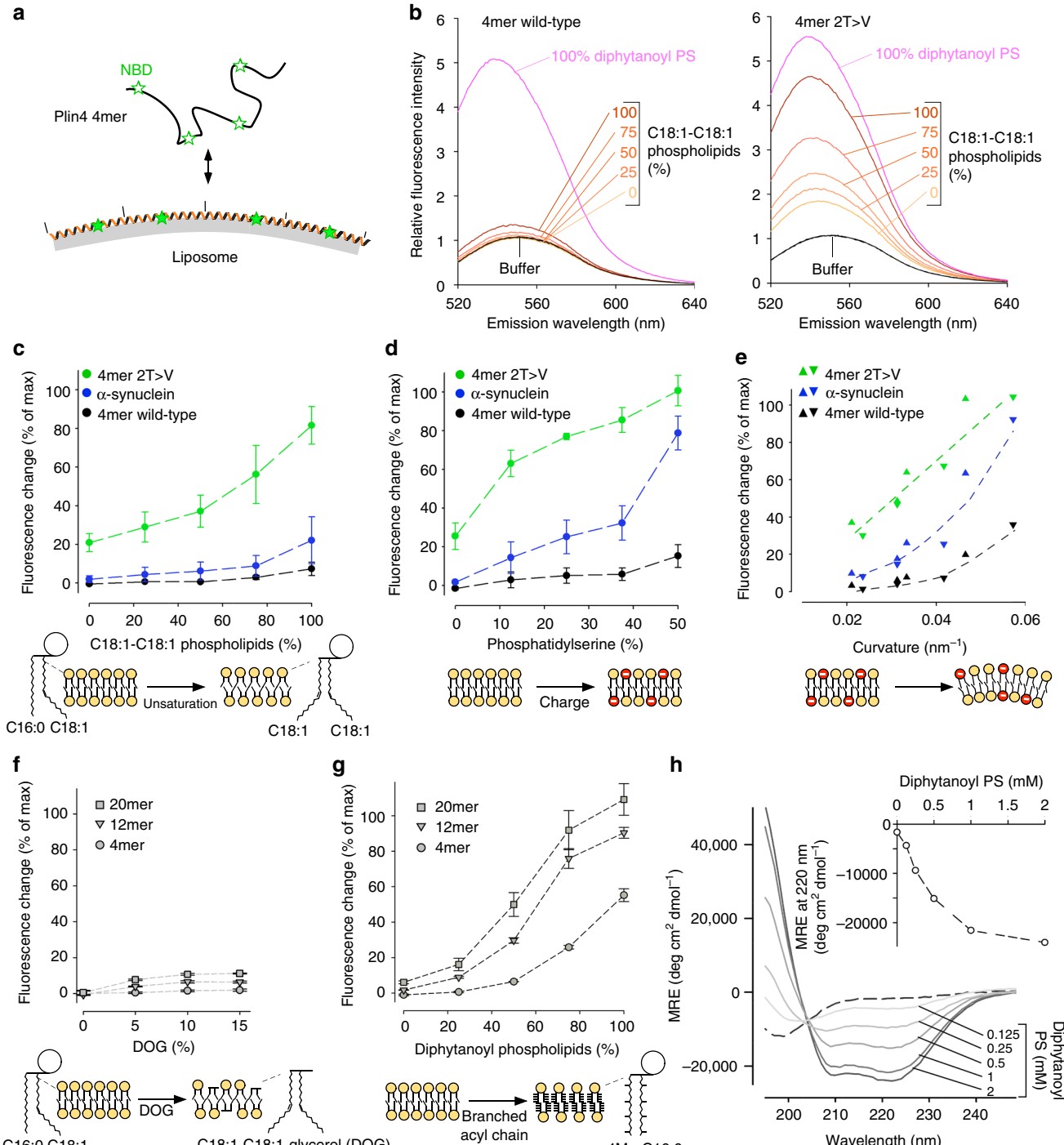

**Fig. 5** Plin4 AH is not adapted to classical phospholipid bilayers. **a** Principle of the assay. Plin4 fragments labeled with NBD were mixed with liposomes and their membrane association was determined by measuring the increase in NBD fluorescence. **b** Typical measurements. NBD-Plin4-4mer or the 2T→V mutant were mixed with liposomes containing PC, phosphatidylethanolamine (PE), and cholesterol (50/17/33 mol%). The acyl chains of PC and PE varied from C16:0-C18:1 to C18:1-C18:1 as indicated. The spectra in solution or with diphytanoyl-PS liposomes are also shown. These spectra were used to calculate % binding for each construct (Methods). **c**–**f** Effect of phospholipid unsaturation, PS, membrane curvature, and DOG on liposome binding of the indicated constructs as determined from experiments similar to **b**. The liposome composition (mol%) was: **c** PC (50), PE (17) and cholesterol (33) with an increasing fraction of C18:1-C18:1 at the expense of C16:0-C18:1 phospholipids; **d** C16:0-C18:1-PE (17), cholesterol (33) and an increasing fraction of C16:0-C18:1-PS (0 to 50) at the expense of C16:0-C18:1-PC (50–0); **e** C16:0-C18:1-PC (30), C16:0-C18:1-PS (20) C16:0-C18:1-PE (17) and cholesterol (33). The liposomes were extruded with polycarbonate filters of decreasing pore radius and the liposome radius was determined by dynamic light scattering; **f** C16:0-C18:1-PE (17), cholesterol (33) and an increasing fraction of DOG (0 to 15) at the expense of C16:0-C18:1-PC (50 to 35). **g** Influence of AH length on binding to diphytanoyl lipids. Liposomes contained PC (50), PE (17), and cholesterol (33) with an increasing fraction of diphytanoyl at the expense of C16:0-C18:1 phospholipids. **h** CD spectra of Plin4-20mer in solution or with increasing amounts of diphytanoyl-PS liposomes. Inset: titration of the CD signal at at 222 nm as a function of liposome concentration. Data shown in **c**, **d**, and **g** are means ± s.e.m. from three independent experiments; **e**–**f** show the results of two independent experiments

d). Particle size did not change with the length of the Plin4 AH (Supplementary Fig. 5d). We also prepared a Plin4-oil emulsion using fluorescently labeled Plin4 and visualized it using a spinning disc microscope. Whereas most particles were below the resolution of the microscope, we could visualize larger oil droplets, which were uniformly coated with fluorescent protein (Fig. 6e).

Many protein emulsifiers interact with neutral lipids by essentially denaturing[58,59]. In contrast, apolipoproteins have been shown to interact with neutral lipids via AHs or β-sheets[10]. Given

the uniform fluorescent signal of Plin4 that we observe around oil droplets, we hypothesize that Plin4 also folds into a secondary structure in contact with oil, possibly an α-helix. Due to high light scattering of the emulsion, we could not test this directly by CD spectroscopy. Instead, we tested if Plin4 displayed any resistance to proteolysis upon triolein binding, which can be an indication of folding[60]. We observed that Plin4 in oil emulsion was more resistant to trypsin than Plin4 in solution (Fig. 6f), suggesting an increase in secondary structure. We note that in the Plin4 emulsion, the majority of the protein (≥90%) was not bound to

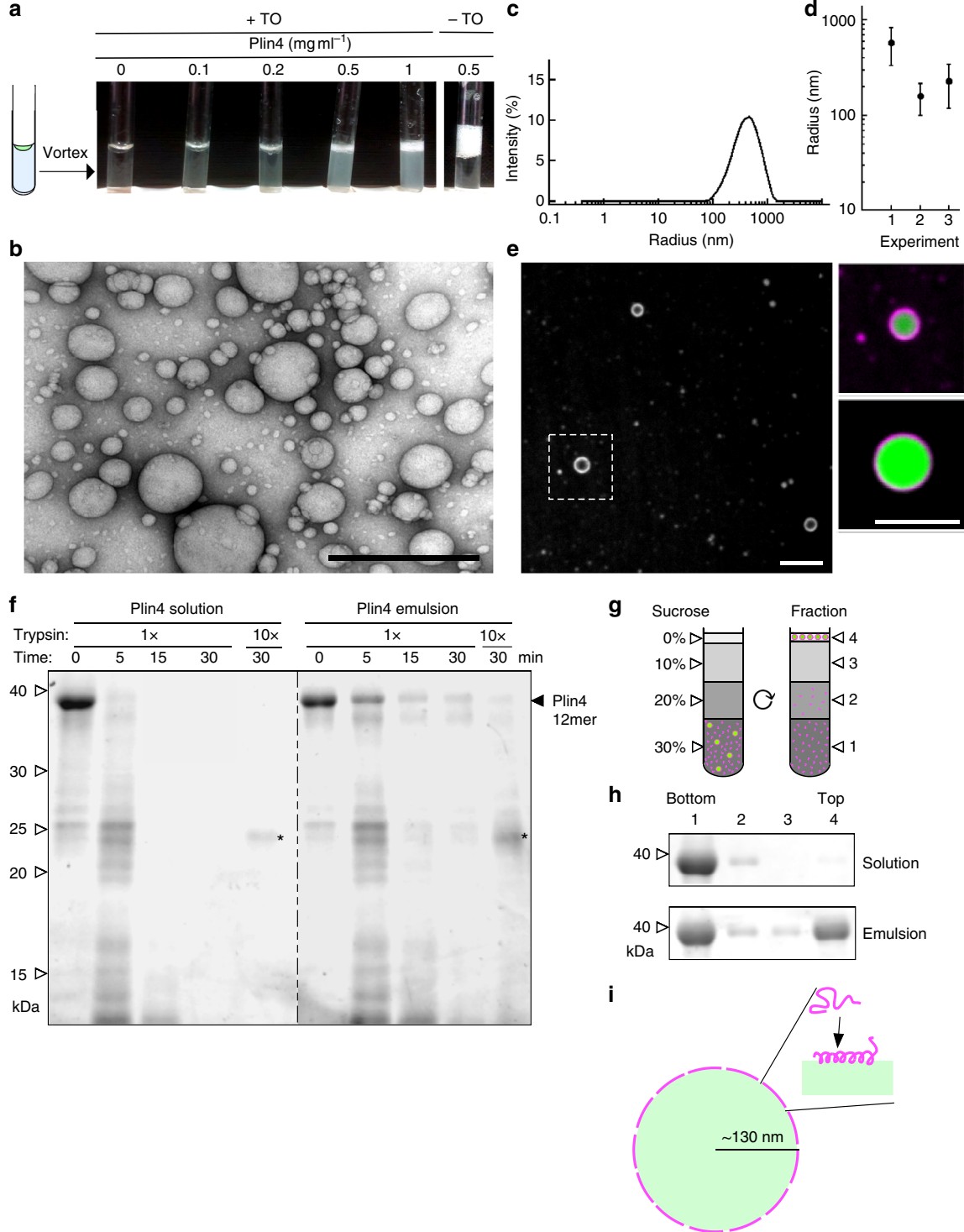

oil, as estimated by flotation of the emulsion through a sucrose gradient (Fig. 6g, h). Therefore, our trypsin assay under-estimates the increase in protease-resistance of Plin4 on oil.

Assuming that Plin4 adopts a perfectly helical conformation in contact with oil, one Plin4-12mer molecule would cover the area of ~60 nm$^2$[38,61]. Further assuming that all triolein in the emulsion experiment (10 µl) was consumed into oil droplets of 200 nm in diameter, this would give us $3 \times 10^{11}$ oil droplets with a total surface area of $1.5 \times 10^{17}$ nm$^2$ (this is an approximation, as we can also see larger drops of oil in the fluorescent-labeled emulsion). If we consider that 10% of Plin4 molecules used in the experiment (=1 nmol) are coating the oil, this gives a total Plin4 helical area of $3.6 \times 10^{16}$ nm$^2$, in the same range as the estimated oil surface area (Fig. 6i; see Supplementary Note 1 for details).

**Plin4 AH expression can rescue the PC depletion phenotype.** The ability of Plin4-AH to directly coat pure triolein suggests that Plin4 could have a protective effect on LDs in cells, notably under conditions where phospholipids are limiting. To test this hypothesis, we turned to Drosophila Schneider 2 (S2) cells, which have been used extensively to study factors that influence LD homeostasis[6,7,62], and where LD production can be strongly induced by exogenous addition of fatty acids (Fig. 7a). We expressed Plin4-12mer in S2 cells as a GFP fusion from an inducible promoter. In the absence of oleic acid, where LDs were largely absent, only soluble signal of Plin4-12mer-GFP was observed in these cells. Upon oleic acid treatment, we observed strong localization of Plin4-AH on LDs (Fig. 7a). Full-length Plin4-12mer is efficiently expressed in S2 cells under all growth conditions (Fig. 7b).

CTP:Phosphocholine Cytidylyltransferase 1 (CCT1), a homolog of human CCTα, catalyzes the rate limiting reaction in the synthesis of PC and has been shown to be particularly important for maintaining stable LDs in S2 cells. When CCT1 is depleted, insufficient PC production leads to a large increase in the size of LDs, decreasing their net surface to volume ratio[6], (Fig. 7c, Supplementary Fig. 6a). We hypothesized that due to the ability of the Plin4 AH to directly coat neutral lipids, its expression under these conditions should decrease the size of LDs. Indeed, the CCT1 depleted cells that were expressing Plin4-12mer-GFP had significantly smaller LDs than non-expressing cells in the same population (Fig. 7c, d). Plin4 expression also led to a small but reproducible decrease in the size of LDs in control cells (Fig. 7c, d; Supplementary Fig. 6b, c). Based on these observations, we conclude that in a cellular context, Plin4-AH can replace the PL monolayer of LDs insufficiently covered with PLs and stabilize them.

## Discussion
We have analyzed the mode of binding of a highly unusual AH, present in the human protein Plin4, to LDs. This AH is exceptional in terms of its length and the repetitiveness of its primary sequence. We demonstrate that the Plin4 AH is exquisitely tuned for binding to LDs and not to bilayer membranes and that this interaction is promoted by depletion of PC from cells, in agreement with the strong interaction of this helix with neutral lipids in vitro. Altogether, these observations suggest that Plin4 acts as a reversible coat that contacts directly the LD core, substituting for phospholipids. By varying the properties of this exceptional AH one at a time, we demonstrate that different AH parameters contribute to specificity and strength of LD binding: length, hydrophobicity, charge and charge distribution. Whereas these properties can to some extent substitute for one another, this may be at the expense of LD selectivity.

Lipid-interacting AHs generally adopt their helical structure when in contact with a lipid surface, whereas in their soluble form they are either unfolded or folded within the soluble conformation of the host protein[16,44,47]. We have demonstrated that the Plin4 AH is unfolded in solution and that it folds into a super long helix in the presence of diphytanoyl liposomes. We show that there is a clear correlation between the Plin4 AH length and its targeting to LDs and to synthetic liposomes (Fig. 1 and Fig. 5). This fits with the "Velcro model", which postulates that the combined effect of low-affinity interactions, repeated over an extended binding surface, leads to the stabilization of the bound AH[15]. Although this model is very intuitive, the previous experimental evidence was less direct or based on small changes in length[10,17,25,63].

A diverse array of AHs has been implicated in LD targeting, and many of them can also bind to lipid bilayers (Table 1). Both α-synuclein, a synaptic vesicle protein, and the nucleo-ER localized enzyme CCTα can be observed on LDs[6,45,64]. Furthermore, the small GTPase Arf1 and the GTP exchange factor GBF1 utilize AHs to target Golgi membranes and also LDs[12,23,62,65]. Interactions of these proteins with LDs are physiologically relevant, as they have reported roles in LD function, with the exception of α-synuclein (but its close homolog γ-synuclein has a function in adipocyte metabolism[46]).

Comparing the AHs that localize to LDs suggests that the LD surface can accommodate a wide range of different AH chemistries. Caution is warranted, as adjacent peptide sequences can influence binding preferences of an AH[66,67]. In contrast, the length and the homogenous nature of the Plin4 AH have allowed us to evaluate the influence of different AH parameters on LD targeting in a context-independent manner. LD targeting is permitted by the weakest AH character, exemplified by the wild-type Plin4 AH sequence (Table 1). This suggest that the LD surface is very permissive for AH binding, in accordance with in silico analysis demonstrating that this surface is abundant in lipid packing defects, which promote AH adsorption[22]. Because packing defects increase with decreasing density of surface phospholipids, conditions that decrease monolayer density may

**Fig. 6** Plin4 interacts directly with neutral lipids in vitro forming oil droplets. **a** Images of tubes in which a drop of triolein (10 µl) was vigorously mixed with a solution (190 µl) of increasing concentration of Plin4-12mer. **b** Representative image of the Plin4-oil emulsion by negative staining electron microscopy. Scale bar: 0.5 µm. **c**, **d** Dynamic light scattering measurement of the size distribution of an aliquot withdrawn from the middle of the oil emulsion obtained with 0.5 mg ml$^{-1}$ Plin4-12mer, and comparison between three independent experiments, with dots representing peak maxima and vertical bars representing polydispersity. **e** Plin4-oil emulsion was visualized by confocal fluorescence microscopy. Unlabeled Plin4-12mer (0.3 mg ml$^{-1}$) was mixed Plin4-12mer-Alexa568 at a ratio 20:1 (magenta), and oil was stained with Bodipy (green). Left panel shows Plin4 and right panels show zoom-ins of merged images. Scale bars: 5 µm. **f** Plin4 in the oil emulsion is protected from degradation by trypsin. Plin4-12mer (1 mg ml$^{-1}$) was incubated in buffer only or vortexed with triolein as in **a**, then digested with 13 µg ml$^{-1}$ (×1) or 130 µg ml$^{-1}$ (×10) trypsin for the amount of time indicated. Samples were analyzed by SDS-PAGE with Sypro Orange staining. Five times less sample was loaded in the 0 min controls than in the other lanes. White arrowheads indicate the migration of molecular weight standards. Asterisks indicate the trypsin band. **g**, **h** Plin4-12mer (1 mg ml$^{-1}$) before (solution) or after (emulsion) the reaction depicted in (**a**) was mixed with sucrose and loaded on the bottom of a sucrose gradient. After centrifugation, four fractions were collected from the bottom and equal volumes were analyzed by SDS-PAGE with Sypro Orange staining. See Supplementary Fig. 7 for uncropped gels in **f**–**h**. **i** Model of a Plin4-12mer-covered oil droplet, drawn to scale. Calculation (see main text) suggests complete coverage of the oil surface by Plin4 AH

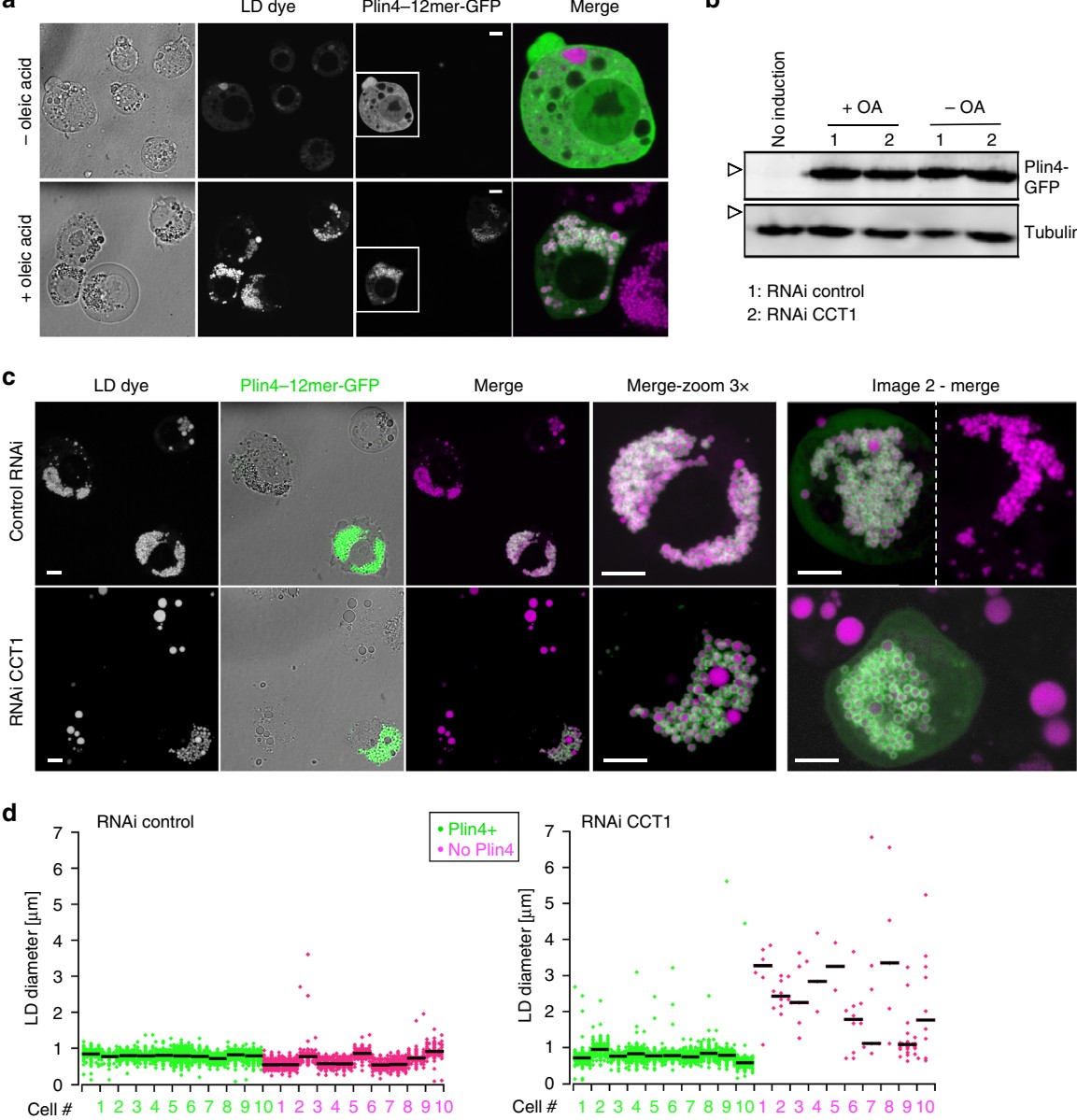

**Fig. 7** Plin4 AH expression in Drosophila S2 cells rescues the defect in LD size due to PC depletion. **a** Plin4-12mer-GFP localizes to LDs. Expression was induced with copper for 48 h. For the last 24 h, cells were either supplemented with oleic acid to induce LDs (+OA) or were kept in normal growth medium (−OA). LDs were stained with Autodot dye. Images were obtained with a spinning disc microscope and are presented as projected z-stacks. The merged images show a ×3 zoom of the area marked in the Plin4 images with LDs in magenta and Plin4 in green. Scale bar: 5 μm. **b** Western blot analysis of Plin4-12mer-GFP expression in non-induced cells (no copper addition, lane 1), or in copper-induced cells (lanes 2–5) that were subjected to the indicated treatments. Plin4-12mer levels were not affected by oleic acid treatment or by RNAi against CCT1 (**c**). Tubulin was used as loading control (uncropped membranes in Supplementary Fig. 7). White arrowheads indicate the migration of 70 kDa and 55 kDa molecular weight standards, respectively. **c**, **d** Cells were treated with RNAi against CCT1 to deplete PC or with control RNAi, followed by Plin4-12mer-GFP induction and oleic acid treatment; **c** images show a comparison of LD size between Plin4-transfected and non-transfected cells in the same field, with enlarged images of the merged channels; **d** graphs show quantification of LD size in individual cells in a representative of three experiments (shown in Supplementary Fig. 6b, c), with bars depicting median values

further promote AH binding, as has been observed[23,24]. The "sticky" nature of the LD surface is also in agreement with findings that LDs can act as sinks for various non-LD proteins to prevent their accumulation in the cytosol[3]. The fact that AHs can easily target LDs suggests that mechanisms that prevent their localization to LDs play a significant role, for example competition between proteins for a limited LD surface[26].

We show that the Plin4 AH can directly coat neutral lipids to make small oil droplets in the absence of any other emulsifier. Perilipins are often described as LD coat proteins, however, this

activity remains poorly characterized[21,27,68]. Various proteins can function as emulsifiers and are in fact widely used for various applications[58]. Many soluble globular proteins bind irreversibly to oils, essentially undergoing denaturation. In contrast, apolipoprotein B, which directly interacts with neutral lipids in low density lipoprotein particles, forms a series of AHs that reversibly bind to lipids in vitro[59].

With its exceptional length, we estimate that one full-length human Plin4 AH of 950 amino acids can replace around 250 phospholipids in a packed monolayer, and could form a surface

that is similarly smooth. The sequence of Plin4 AH, with its high repetitiveness, low hydrophobicity and particular charge organization, may be exquisitely adapted to forming a homogeneous coat whose recruitment to LDs is highly reversible. It is interesting to note a parallel with the structure of synthetic peptoid nanosheets that assemble on an oil–water interface and are stabilized by side-side electrostatic interactions[69]. Further studies will be needed to understand the structural organization of the Plin4 coat.

If Plin4 AH can directly interact with neutral lipids in vitro, what is the situation in cells? Branched diphytanoyl phospholipids also promote binding of Plin4 AH to liposomes, suggesting that the presence of polar lipids does not preclude binding. Instead, the Plin4 AH seems to bind to any surface with a very high density of packing defects, for example an LD with increased surface tension. Although Plin4 is highly expressed in adipocytes, and its unusual structure suggests a specialized function, little is known about the physiological role of this protein[29]. Our data suggest that Plin4 may be optimized for coating and thereby stabilizing LDs under conditions of limiting phospholipids. Interestingly, Plin4 has been observed to preferentially localize to small LDs in differentiating adipocytes[33,34], suggesting a function in regulating LD surface to volume ratio. In addition, it has been shown to display a preference for LDs with a particular neutral lipid composition[43]. Further studies are needed to analyze the properties and dynamic behavior of this protein coat in vitro and in cells. The AH of Plin4 is about 10 times longer than the longest predicted AH region in other human perilipins, in which the repeats are less conserved and sometimes broken up by deletions or insertions. However, compared to AHs found in other proteins, these AH regions are all still very long, and it will be interesting to explore to what extent our findings on Plin4 AH apply to perilipins in general.

Finally, we note that a study published while this paper was under review also concludes that the LD surface can interact with AHs of widely different chemistries[70]. However, the authors suggest that one requirement for LD binding is the presence of large hydrophobic residues in the AH, which is at odds with our results based on Plin4. Because our study indicates that increasing the size of hydrophobic residues promotes a general increase in the binding of AHs to both LDs and bilayer membranes, such a mechanism probably applies to proteins like Arf that have a short AH and act on both LDs and on bilayer-bound organelles[62,65]. For proteins like perilipins, AH length and electrostatic interactions compensate for the lack of large hydrophobic residues and lead to a more selective mode of LD binding.

## Methods

**Sequence analysis**. Sequences of AHs were identified and plotted using Heliquest[71]. Heliquest was also used to calculate their hydrophobicity and hydrophobic moment. For longer AHs containing more than 3 helical turns, hydrophobic moment was calculated as the average of values for individual helices, as shown in figures, normalizing for the length of AHs. The amino acid conservation of the 33 amino acid repeats of Plin4 was analyzed using Weblogo[72].

**Plasmid DNA construction**. All human Plin4 expression plasmids used in this study (Supplementary Table 3) were constructed using synthetic double-stranded DNA fragments. Due to the highly repetitive nature of the Plin4 AH region, DNA sequences were optimized for synthesis using the algorithm on the Eurofins website (https://www.eurofinsgenomics.eu). Plin4-4mer, Plin4-8mer, and Plin4-12mer–mCherry fusion plasmids for expression in HeLa cells were generated by cloning synthetic genes (Supplementary Table 4) into the pmCherry-N1 expression plasmid (Clontech) using XhoI and BamHI restriction sites. Note that we define a 1mer unit as 33 amino acids of the Plin4 AH sequence. Plin4-2mer was generated by PCR-amplification of the first 67 codons of Plin4-4mer and XhoI-BamHI into pmCherry-N1. Plin4-20mer was generated by assembling synthetic genes for Plin4-8mer and Plin4-12mer into pmCherry-N1 using GeneArt Assembly kit (Life Technologies), without introducing any additional nucleotides between the DNA sequences encoding the two Plin4 fragments. The C-terminal region of Plin4,

encoding amino acids 1060-1356 of mouse Plin4, was PCR-amplified from vector pKTD-16A (gift from Knut Dalen, University of Oslo) and cloned XhoI-BamHI into vectors pmCherry-N1 and pEGFP-N1. Plin4 mutants were generated using synthetic genes whose sequences were optimized for DNA synthesis. All 4mer mutants were exact 4× repeats of a 33 amino acid sequence, based on the parental sequence of human Plin4 fragment aa246-278 (Supplementary Table 4). All mutant synthetic genes were cloned BamH1-XhoI into pmCherry-N1.

For expression of Plin4 constructs in *Saccharomyces cerevisiae*, the pRS416-derived CEN plasmid pRHT140 (gift from Sebastien Leon, IJM, Paris) containing ADH1promotor and GFP expression tag for C-terminal tagging, was mutated by site-directed mutagenesis to introduce an Nhe1 restriction site and correct the reading frame, generating pKE28. DNA sequences encoding various Plin4 fragments were then subcloned by restriction and ligation from pmCherry-N1 plasmids using HindIII and NheI restriction sites. For expression of Plin4 constructs in *E. coli* without any tag, various Plin4 fragments were PCR-amplified to introduce NheI and XhoI restriction sites and ligated into pET21b (Novagen), resulting in the elimination of the T7 and His tags from the vector. The only exception was plasmid pKE25 for expression of Plin4-20mer, which was generated by restriction and ligation of the Plin4-20mer DNA fragment from the pmCherry plasmid pCLG70 using NheI and BamHI restriction sites. For this purpose, pET21b was mutagenized by site-directed mutagenesis to include 3 stop codons after the BamHI restriction site in the vector, resulting in pET21b-3stop. For expression of Plin4-12mer in Drosophila S2 cells, Puro-pMTWG vector was generated by digestion of pAWG destination vector (the Drosophila GatewayTM Vector Collection) by EcoRV and BglII enzymes to excise the actin promoter. Puromycin resistance gene and metallothionein promoter were obtained by PCR from pMT-Puro vector (Riken, ref. RDB08532), and subsequently inserted by In-Fusion reaction (Clontech) into the destination vector. Primers used are TCATTTTTCCAGATCTCGGTACCCGATCCAGACATGATAAG (Fw); TAGACAGGCCTCGATATCCCTTTAGTTGCACTGAGATGATTC (Rv). DNA for Plin4-12mer was PCR-amplified and cloned into this vector using the Gateway LR reaction technology (Life Technologies).

**Yeast growth and media**. Yeast strains used were BY4742 MATα his3Δ1 leu2Δ0 lys2Δ0 ura3Δ0, or into BY4742 ERG6::mRFP::KanMX6. Yeast cells were grown in yeast extract/peptone/glucose (YPD) rich medium, or in synthetic complete (SC) medium containing 2% glucose. Yeast was transformed by standard lithium acetate/polyethylene glycol procedure. For observation of LDs, liquid cell cultures were inoculated from a single colony and grown for 24 h at 30 °C in SC-Ura with 2% glucose to maintain plasmid selection.

**Cell culture and transfection**. HeLa cells were grown in Dulbecco's modified Eagle's medium (DMEM) supplemented with 4.5 g l$^{-1}$ glucose (Life technologies), 10% fetal bovine serum (FBS, Life technology) and 1% Penicillin/Streptomycin antibiotics (Life technologies). For protein expression, subconfluent cells were transfected with Lipofectamine 2000 (Invitrogen) in Optimem medium (Life technologies) for 6 h, followed by 16 h in standard growth medium or standard growth medium containing 250 µM oleic acid (Sigma) in complex with fatty-acid free BSA (Sigma).

Drosophila S2 cells (ThermoFisher) were cultured in Schneider's Drosophila medium (Invitrogen) supplemented with 10% FBS and 1% Penicillin/Streptomycin at 25 °C. For generating stably-transfected cells, cells were incubated with plasmid DNA and TransIT-Insect Reagent (Mirus), followed by selection with 2 µg ml$^{-1}$ puromycin (Life technologies) for 2 weeks. Protein expression from the metal-inducible promoter was induced for 48 h with the addition of 100 µM Cu-sulfate to the medium. Lipid droplets were induced with 1 mM oleic acid-BSA complex for 24 h.

**RNAi depletion**. RNAi depletion against CCT1 in stably-transfected S2 cells was performed as described[6], using the same primer sequences to generate 580 bp dsRNA (Fwd, ACATCTATGCTCCTCTCAAGG C; Rev, CTCTGCA-GACTCTGGTAACTGC). For RNAi control experiment, we used dsRNA against luciferase (Fwd, AAATCATTCCGGATACTGCG; Rev, CTCTCTGAT-TAACGCCCAGC). The dsRNA fragments were generated with T7 RiboMAX™ Express RNAi System (Promega) using two separate PCR reactions with a single T7 promoter (sequence TAATACGACTCACTATAGG appended to 5′ ends of primers). 2 × 10$^6$ cells were incubated with 30 µg of dsRNA for 30 min in serum-free medium, followed by the addition of complete medium. Cells were incubated for 3 days. Protein expression was induced with Cu-sulfate for the last 2 days, and LDs were induced with 1 mM oleic acid during the final day of the experiment. To verify the depletion of CCT1, total RNA was prepared using the NucleoSpin RNA kit (Macherey-Nagel), and quantiTect Reverse Transcription kit (Qiagen) was used to synthesize complementary DNA. Quantitative real-time PCR was performed in triplicates using Light Cycler 480 SYBER Green I Master Kit (Roche) with primers for CCT1 (Fwd, GGAAGCGGACCTACGAGATA; Rev, GTGCCCTGATCCT-GAACTT), and for GAPDH as control (Fwd, ATGAAGGTGGTCTCCAACGC; Rev, TCATCAGACCCTCGACGA).

**Western blot analysis**. Total cell lysates from S2 cells were obtained by incubating harvested cells in ice-cold lysis buffer (300 mM NaCl, 100 mM Tris-HCl pH 8, 2% NP-40, 1% deoxycholate, 0.2% SDS, 10 mM EDTA, 'Complete mix' protease inhibitors from Roche), followed by centrifugation and denaturation in Laemmli buffer at 95 °C for 5 min. Proteins were separated on 13% SDS-PAGE gel and transferred to a nitrocellulose membrane (GE Healthcare). Rabbit polyclonal αGFP antibody (Invitrogen, A11122, 1:2000 dilution) was used for detection of Plin4-12mer-GFP, followed by HRP-conjugated secondary antibody. Mouse monoclonal α-tubulin (Sigma T9026, 1:4000 dilution) was used for loading control. ECL Western Blotting kit (GE Healthcare) was used for detection, and images were obtained using Fujifilm LAS3000.

**Fluorescent microscopy**. Transfected HeLa cells were fixed with 3.2% paraformaldehyde (Sigma) in PBS for 30 min at room temperature. After washing three times with PBS, cells were stained with Bodipy 493/503 (Life Technologies) at 1 µg ml$^{-1}$ for 30 min at room temperature and washed three times with PBS. Cells were mounted on coverslips with Prolong (Life technologies). Images were acquired with a TCS SP5 confocal microscope (Leica) using a ×63/1.4 oil immersion objective driven by LAS AF Lite software. Alternatively, images were acquired with an LSM780 confocal microscope (Zeiss) using a ×63/1.4 oil immersion objective and a PMT GaAsP camera, driven by ZEN software. Mid-focal plane images were selected and they were processed with ImageJ and with Canvas Draw software. Figures were compiled in Canvas Draw.

Yeast cells were grown to post-diauxic shift (24 h) in selective media and imaged directly on glass slides. Images were acquired at room temperature with a DMI8 (Leica) microscope, equipped with an oil immersion plan apochromat 100 objective NA 1.4, an sCMOS Orca-Flash 4 V2+ camera (Hamamatsu), and a spinning-disk confocal system CSU-W1 (Yokogawa - Andor) driven by MetaMorph software. 5 z-sections separated by 0.5 µm were acquired. For quantification of plasma membrane and LD signal in yeast, images were acquired with LSM780 confocal microscope as described above. To visualize LDs and stably-transfected proteins in Drosophila S2 cells, cells were stained with Autodot autophagy visualization dye (Cliniscience) for 1 h and washed two times with PBS, after which they were resuspended in fresh growth medium and imaged directly on untreated glass slides using the CSU-W1 spinning-disk set-up described above. For 3D-reconstitutions of S2 cells, between 16 and 25 z-sections separated by 0.2 µm were acquired for each image.

**Image analysis**. Images of HeLa and yeast cells were analyzed using ImageJ. To determine the fraction of LDs in HeLa cells that were positive for transfected fluorescent protein, a single z-section that contained the most LDs in a cell was first selected. All LDs in the selected cell section were identified in the green (Bodipy dye) channel using the 'Analyze particle' plug-in. The total area of LD fluorescent signal in the cell was also measured, and the average size of LDs was determined by dividing these two values. LDs positive for fluorescent protein were then identified by determining a threshold value for the red fluorescent signal (mCherry-protein fusion), ×1.4 above average cellular fluorescence, and counting all LDs with fluorescence above this threshold. This number was divided by the total LD number to calculate the fraction of LDs in one cell section positive for protein. To determine the maximum intensity of fluorescent protein signal on LDs, five brightest LDs per cell were selected manually and the maximum fluorescent signal was determined from a line-based profile using the 'Find peaks' plug-in. The average of these five maxima was taken as the maximum LD fluorescence intensity. Similarly, the maximum cytoplasmic fluorescence was determined from the five maxima of line-based profiles going from the nucleus to the cell exterior, with lines drawn manually and not crossing any LDs. The average protein fluorescence in the nuclear area was determined by placing three squares in the nucleus of each cell and calculating their average fluorescence. Data was processed in Microsoft Excel.

To determine the diameter of LDs in Drosophila S2 cells, 3D-representations of S2 cells were generated from z-stacks and LDs were identified using spot detection with Imaris image analysis software (Bitplane, Oxford). Diameters of all spheres representing single LDs were recorded. LDs that were very close together and could not be identified by the software as individual spheres were manually eliminated. Therefore, the total number of LDs identified in each cell is lower than the actual number. LDs in all isolated cells transfected and non-transfected in a selected image were quantified. Data were processed in Excel and in KaleidaGraph.

**Protein purification**. Purification and labeling of full-length α-synuclein were as previously described[17]. The protocol for purifying the various Plin4 constructs was adapted from that used for α-synuclein[50], and included two steps: boiling to precipitate most proteins except Plin4, which is heat resistant, followed by ion exchange chromatography. In contrast to α-synuclein, Plin4-AH has a net positive charge at neutral pH (pI = 9.56 for the amphipathic region [95–985]). Therefore, Plin4 fragments were purified by cation chromatography. E. coli cells BL21DE3 transformed with expression plasmids were grown to O.D. ≈0.6 and induced with 1 mM IPTG for 3–4 h at 37 °C. Cells from 0.25 to 0.5 l cultures were collected by centrifugation and frozen. The bacterial pellet was thawed in 25 mM Tris, pH 7.5, 120 mM NaCl, supplemented with 1 mM DTT, 0.1 mM PMSF, and other protease inhibitors (Roche complete cocktail, phosphoramidon, pepstatin, bestatin). Cells were broken by three passages in a french press or by sonication. The lysate was

centrifuged at 100,000 × g for 30 min at 4 °C in a TI45 rotor (Beckman). The supernatant in centrifuge tubes was immersed in boiling water for 30 min. The resulting cloudy suspension was centrifuged at 100,000 × g for 15 min at 4 °C to remove precipitated material. The supernatant was dialyzed against 25 mM Tris, pH 7.5, 10 mM NaCl, 1 mM DTT for 2 h in a cold room using Spectra/Por membranes with a cut off of 6000 Da (Spectrum labs) and then centrifuged again at 100,000 × g for 20 min at 4 °C. For chromatography, the final supernatant was loaded onto a Source S column (7.5 ml; GE Healthcare) and submitted to a salt gradient from 1 mM to 400 mM NaCl (8 column volumes) at a flow rate of 4 ml min$^{-1}$ using an Akta purifier system (GE Healthcare). All Plin4 constructs (4mer, 12mer, 20mer, 4mer 2T→V mutants), eluted at approximately 100 mM NaCl. Alternatively, the purification was performed using a 1 ml hand-driven column (HiTrap, GE Healthcare) using a step salt gradient. After SDS-PAGE analysis of the chromatography fractions, the protein pool was divided in small aliquots, flash frozen in liquid nitrogen and stored at –80 °C. Of note, the lack of aromatic residues in Plin4-AH prevents the determination of its concentration by UV spectroscopy or by Coomassie Blue-based assays (e.g. Bradford). As a first estimation, we loaded the various Plin4 aliquots on SDS-PAGE along with BSA standards and stained the gel with Sypro Orange (Life Technologies), which interacts with SDS. This procedure gave a typical Plin4 concentration in the range of 1–2 mg ml$^{-1}$, except in the case of the 2T→V mutant, where the purification yield was lower (concentration ≈0.3 mg ml$^{-1}$). We also used a more accurate protocol of determining the protein concentration through cysteine quantification with the Edman reagent (Sigma), relying on numerous cysteines in Plin4-4mer, 12mer and 20mer constructs (4, 4, and 10, respectively). This procedure was performed after protein dialysis to eliminate DTT and gave a 2.5 higher concentration of protein for the wild-type constructs (in the range of 3.5 mg ml$^{-1}$).

**Analytical gel filtration**. Plin4-4mer, 12mer and 20mer were analyzed by gel filtration on a Superose-12 column (10×300 mm, GE Healthcare). Proteins were injected at a starting concentration and volume of 3.5 mg ml$^{-1}$ and 100 µl. Elution was performed at room temperature in 25 mM Tris, pH 7.7, 125 mM NaCl, 0.5 mM DTT at a flow rate of 0.5 ml min$^{-1}$. Absorbance was continuously measured at 215 nm. Fractions were collected and analyzed by SDS-PAGE using Sypro Orange staining. To calibrate the column, the following standards were used: cytochrome C (MW: 12.4 kDa, Stoke's radius: 17 Å), Anhydrase Carbonic (29 kDa, 20 Å), Bovine Serum Albumin (67 kDa, 35 Å), ß amylase (200 kDa, 54 Å) and Ferritin (443 kDa, 61 Å).

**Circular dichroism**. The experiments were done on a Jasco J-815 spectrometer at room temperature with a quartz cell of 0.05 cm path length. Each spectrum is the average of several scans recorded from 200 to 260 nm with a bandwidth of 1 nm, a step size of 0.5 nm and a scan speed of 50 nm min$^{-1}$. Control spectra of buffer with or without liposomes were subtracted from the protein spectra. The buffer used was Tris 10 mM, pH 7.5, KCl 150 mM.

**Protein labeling with fluorescent probes**. Purified Plin4 constructs were labeled with fluorescent probes via endogenous cysteines. For NBD (nitrobenzoxadiazole) labeling, 1 ml of Plin4-4mer, 12mer, 20mer, or 2T→V 4mer mutant was dialyzed on a NAP10 column in Hepes 50 mM K-Acetate 120 mM, pH 7.4 (HK buffer) to remove DTT. The protein fraction was then incubated for 5 min at room temperature with 8–20 mol excess of NBD-iodoacetamide (Life Technologies), corresponding to a 2 mol excess over cysteines. After the addition of 2–5 mM DTT to stop the reaction, the mixture was loaded again onto a NAP10 column to separate the labeled protein from the excess of probe. The fractions were analyzed by SDS-PAGE and UV-visible chromatography. For labeling with Alexa C5 maleimide probes (Life Technologies), we used a similar protocol except that the probe was incubated with the protein at 1:1 ratio for 5 min at 4 °C before the addition of DTT to stop the reaction.

**NBD fluorescence assay for liposome binding**. Dry films containing the desired amount of phospholipids and cholesterol were prepared from stock solutions of lipids in chloroform (Avanti Polar Lipids) using a rotary evaporator. The film was resuspended in 50 mM Hepes, pH 7.2, 120 mM K-acetate at a concentration of 2 mM phospholipids. After five cycle of freezing in liquid nitrogen and thawing in a water bath at 30 °C, the resulting multi-lamellar liposome suspension was extruded through polycarbonate filters. Filters with a pore size of 100 nm were used for all experiments except when studying the effect of membrane curvature. In this case, the liposome suspension was sequentially extruded through 200, 100, 50, and 30 nm pore size filters, or sonicated with a titanium probe to get the smallest liposomes. The size distribution of the liposomes was determined by dynamic light scattering using a Dynapro apparatus.

Fluorescence emission spectra upon excitation at 505 nm were recorded in a Jasco RF-8300 apparatus. The sample (600 µl) was prepared in a cylindrical quartz cuvette containing liposomes (150 µM lipids) in HK buffer supplemented with 1 mM MgCl$_2$ and 1 mM DTT. The solution was stirred with a magnetic bar and the temperature of the cuvette holder was set at 37 °C. After acquiring a blank spectrum, the protein was added and a second spectrum was determined and corrected for the blank. The Plin4 4mer, 12mer, and 20mer constructs were used at 120, 40, and 24 nM, respectively, in order to have the same concentration of amino

acids (≈15 μM) and thus the same amino acid/phospholipid ratio (≈1:10) in all experiments. The fraction of liposome bound protein was defined as $(F−F_{sol})/(F_{max}−F_{sol})$, where $F$ is the actual fluorescence level at 540 nm, and $F_{sol}$ and $F_{max}$ are the fluorescence levels of the NBD-labeled construct in buffer and in the presence of diphytanoyl-PS liposomes, respectively. Flotation experiments in the presence of diphytanoyl-PS liposomes indicated that all Plin4 constructs as well as α-synuclein were completely bound to these liposomes.

**Preparation and analysis of protein-oil emulsions.** Dilutions of purified Plin4-12mer (up to 1 mg ml$^{-1}$, or 25 μM) were prepared in HK buffer supplemented with 1 mM MgCl$_2$ and 1 mM DTT. 190 μl of each solution was pipetted into a 600 μl glass tube, and a 10 μl drop of triolein (Sigma) was added to the top. They were vortexed for three cycles of 30 s on 30 s off at 25 °C under argon atmosphere. Images of resulting emulsions were taken with a phone camera. Measurements of the mean hydrodynamic radius of the Plin4-oil droplets by dynamic light scattering were performed on a sample taken from the middle of the tube, avoiding any un-reacted oil that remained at the top of the emulsion, at least 3 h after vortexing to prevent the interference of gas bubbles with the measurement. Measurements were performed on a Zetasizer nano ZS machine (Malvern) at 25 °C, and data were processed using the CONTIN method.

For analysis by electron microscopy, samples of emulsion were deposited on glow discharge carbon-coated grids and negatively stained with 1% uranyl acetate. They were observed with a JEOL 1400 transmission electron microscope using a Morada Olympus CCD camera.

For analysis by fluorescent microscopy, emulsions were prepared as described, except that Plin4-12mer was mixed with Alexa-568-labeled Plin4-12mer at a ratio 20:1, and then mixed with triolein containing 2 μg ml$^{-1}$ Bodipy 493/503. Fluorescent oil droplets were visualized directly on untreated glass coverslips with the spinning-disk confocal system. 10 to 15 z-sections of 0.2 μm were acquired in order to visualize both small and large droplets.

**Separation of Plin4-oil emulsion on sucrose gradients.** The Plin4-oil emulsion was prepared as above using 1 mg ml$^{-1}$ Plin4-12mer. Next, 60% w/v solution of sucrose in HK buffer supplemented with 1 mM MgCl$_2$ and 1 mM DTT was added to the emulsion, to obtain a final sucrose concentration of 30%. 450 μl of the resulting suspension was loaded on the bottom of a centrifuge tube and overlaid with a step sucrose gradient consisting of 300 μl 20% w/v sucrose, 300 μl 10% w/v sucrose, and finally with 100 μl of buffer alone. The samples were centrifuged at 50000 rpm (214,000 × $g$) in a Beckman swing-out rotor (TLS 55) for 80 min at 8 °C. Four fractions were carefully collected from the bottom with a Hamilton syringe, having the following volumes: 450 μl, 300 μl, 320 μl, and 80 μl, respectively. Equal volumes of all fractions were analyzed by SDS-PAGE and proteins were stained with Sypro Orange (Life Technologies). The gels were imaged with a FUJI LAS-3000 fluorescence imaging system.

**Trypsin protection assays.** Plin4-oil emulsion was prepared using Plin4-12mer (1 mg ml$^{-1}$) and triolein as described above. At time zero, 100 μl of this emulsion or of Plin4-12mer starting solution were mixed with 13 or 130 μg ml$^{-1}$ trypsin (Sigma) solution. At the indicated times, 30 μl of samples were withdrawn and added 3 μl of 100 mM PMSF (Sigma) to stop the reactions, then stored on ice. Reactions were analyzed by SDS-PAGE and Sypro Orange staining.

**Data availability.** All plasmids, DNA sequences and bacterial or yeast strains used in this study are available upon request (see Supplementary Tables 3 and 4). All relevant data are available from the authors.

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

## Acknowledgements

We thank Karine Eudes and Joel Polidori for technical support, Romain Gautier for help with bioinformatic analyses, the ImagoSeine facility at the Institut Jacques Monod, member of IBiSA and the France-BioImaging (ANR-10-INBS-04) infrastructure, and in particular Xavier Baudin and Vincent Contremoulins for help with imaging and image analysis, Florent Carn for assistance with dynamic light scattering, Sebastien Léon for yeast plasmids and reagents, and Jean-Marc Verbavatz, Veronique Albanèse, Luc Bousset, Yvon Jaillais and members of our labs for comments and discussions. This work was supported by the French National Research Agency (ANR), grant LDsurfDynamics (ANR-13-BSV2-0013) to C.J. and B.A., "Fondation pour la Recherche Médicale", grant number DEQ20150934717 to C.J. and A.C., Marie Curie career integration grant (631997) to A.C., PhD fellowship from the French "Ministère de l'Education National, de l'Enseignement Supérieur de la Recherche" to M.G.-A., and postdoctoral fellowship from the Basque Foundation for Science (Ikerbasque) to M.M.

## Author contributions

A.C. designed and supervised research. A.C., S.A.-B., M.G.-A., and C.L.T.G. performed the cell biology experiments. M.G.-A., M.M.M., A.C., and B.A. performed the in vitro experiments. S.P. and B.A. performed electron microscopy. A.C., C.L.T.G., S.A.-B., M.G.-A., M.M., and B.A. analyzed data. J.G. contributed unpublished reagents. B.A. and C.L.J. participated in the initiation of the project and assisted with the manuscript. A.C. wrote the manuscript.

## Additional information

**Competing interests:** The authors declare no competing interests.

