## [Peer Review File · Nature Communications]

PEER REVIEW FILE

Reviewers' comments:

Reviewer #1 (Remarks to the Author):

The work by Copic et al. describes work to delineate the mechanism by which LD binding proteins selective bind to the LD surface. As model protein to use perilipin 4, which, when compared to all other perilipins, is highly unusual in its domain structure consisting predominantly of a repeating stretch of aa that may form an amphipathic alpha helix. The exact structure of this protein is currently unknown, as are its interactions with lipids. This work significantly contributes to our understanding of this important unresolved area of LD biology. That being said I do have some significant comments and suggestions for edits.

1. For of all the authors use the word “outstanding” to describe the alpha helix of perilipin 4 (in the abstract and throughout their paper). I do not like this terminology as it can easily be misinterpreted. The authors should change this word to either “unusual” or “exceptional in length and xxx”. Again, just the word outstanding in English has different meanings and may lead to misinterpretation of the authors meaning. Exceptional is better but then clarify in the text exactly in what way it is exceptional. Unusual is perhaps the least controversial replacement.

2. Line 63. This sentence at the end of the second paragraph on page 3 is incomplete.

3. In the bottom paragraph of page 3 the authors refer to work done by the authors themselves and others on the interaction of AH domains with bilayer membranes. However, there has been extensive work performed by others on the interaction of AH domains with lipid monolayers. Specifically work on apolipoproteins but also, more recently, on perilipins. The authors need to cite this work and comment on how this work relates to their current study. Specifically the following two references are currently lacking in this work and need to be recognized:

Sletten A, Seline A, Rudd A, Logsdon M, Listenberger LL. Surface features of the lipid droplet mediate perilipin 2 localization. *Biochem Biophys Res Commun*. 2014 Sep 26;452(3):422-7. doi: 10.1016/j.bbrc.2014.08.097.

and

Mirheydari M, Rathnayake SS, Frederick H, Arhar T, Mann EK, Cocklin S, Kooijman EE. Insertion of perilipin 3 into a glycerol(phospho)lipid monolayer depends on lipid headgroup and acyl chain species. *J Lipid Res.* 2016 Aug;57(8):1465-76. doi: 10.1194/jlr.M068205.

Specifically this latter work discusses some aspects of perilipin interaction with LDs that need to be discussed in the current manuscript. The comment in this paragraph “It is not known which parameters are important for AH binding to the LD surface.” is thus not entirely correct. Additionally there are other works showing how ER resident proteins end up on LDs. E.g. caveolin, and this work is also relevant for this paper.

In short the discussion, while good, needs some expansion to incorporate what is known.

Results section:

4. The authors nicely discuss and show the aa sequence and predicted 3-11 structure of perilipin 4 and compare this to other perilipins. The figure in the supplementary results is hard to read though. Please enhance the resolution. Currently it is not acceptable for publication.

5. The authors use AH-GFP fusion proteins for their cellular work to determine LD binding. While the data seem convincing, I wonder if the fusion proteins themselves have a strong effect on LD / membrane binding? In other words the authors do not show if their constructs are soluble by themselves or if they form intracellular aggregates when not fused to GFP or another fluorescent protein. I thus miss a convincing control.

5. On line 150 the authors suggest there are no yeast perilipins. This is in fact incorrect. Here is a paper the authors may want to consult: Gao Q, Binns DD, Kinch LN, Grishin NV, Ortiz N, Chen X, Goodman JM. Pet10p is a yeast perilipin that stabilizes lipid droplets and promotes their assembly. *J Cell Biol.* 2017 Aug 11. pii: jcb.201610013. doi: 10.1083/jcb.201610013

6. On page 9 the authors address the issue of charge for protein targeting. This issue appears to be highly complex and authors make some interesting points concerning protein and lipid charge in this respect. While I agree with the authors on most aspects of this point (the fact that its complex) I do feel the authors are missing part of the story. First is the observation that oil droplets by themselves are negatively charged in water (or any aqueous solution, see for example:

Ghimire C, Koirala D, Mathis MB, Kooijman EE, Mao H. Controlled particle collision leads to direct observation of docking and fusion of lipid droplets in an optical trap. *Langmuir.* 2014 Feb

and references herein). Additionally, the acidic amino acid residues do not necessarily remain negatively charged when interacting with a lipid surface (or any surface). In fact the pKa of these residues depends sensitively on local conditions. While this may not be of effect here, I think it is important the authors consider these possibilities.

7. Top of page 10. Here the authors further discuss their results on their experiments with charged residues of perilipin 4. It would be nice to see a further discussion (here or in the discussion) on the relevant physics of lipid interfaces which is important but which the authors skip over. See my previous comment as well.

8. The authors (on page 10 in section of liposome interactions) discuss work with lipids of different saturation. Others have previously shown that this is indeed important for perilipin interaction with lipid interfaces, but not in the way the authors postulate here (unsaturation perturbs perilipin 3 insertion, not increase it). See the reference (Mirheydari et al) given above for further details and additional references. The authors need to cite these works in their manuscript and comment on the results in relationship to these previously published works.

9. I do not understand why the authors have not used diacylglycerol in their liposome experiments. While DAGs can cause issues with liposome formation, small amounts can easily be incorporated when care in liposome preparation are taken. The authors have made a big deal about the curvature defects generated by DAG in the past so I was highly surprised to see work on a non-natural lipid in this regard. While the results with diphytanoyl lipids are certainly interesting they are perhaps not physiologically relevant. The authors need to include data on diacylglycerol, a lipid likely present in the lipid monolayer surrounding LDs, or at least justify in the paper why they did not use DAG. There is experimental evidence (from Nate Wollins group) that shows that DAG is the binding partner for perilipin 3.

10. I am worried about the work with pure triolein that the authors present starting on page 12. While they do some controls and mention protein denaturation, they do NOT convincingly show that perilipin 4 is not denatured by their vortexing procedure. There are far better experiments that one can do that are convincing and preclude the possibility of protein denaturation. This section should be removed from the manuscript. So, the authors make a case for, but do not provide convincing proof, that denaturation is not a problem in these experiments.

Discussion:

11. First paragraph. The authors need to be more careful when they interpret their results obtained from their oil experiments. These results are too preliminary and should be removed.

The remainder of the paper is sufficient to be published without this.

12. line 453. The authors need to cite the works mentioned above. And discuss their results in light of these previous works.

13. Lines 484-486. This statement is preliminary and needs to be evaluated in light of the biophysics of the LD lipid interface. Some important aspects of which the authors do not mention in this paper. Which, of course, is ok as this is not their main focus. BUT they need to write this particular statement more carefully in light of other results and what is known of lipid interface electrostatics.

14. paragraph 2 on page 17. This should be removed or extensively rewritten. See my comments above.

15. paragraph 3 on page 17. There are numerous papers from a Mexican group on the interaction of apolipoproteins and their peptides with lipid interfaces. This work may be relevant for the types of interactions the authors suggest here. The main authors of these papers are Xicohtencatl-Cortes, Mas-Oliva and Castillo.

Overall this is exciting new work and deserving of publication after significant revision.

Reviewer #2 (Remarks to the Author):

This manuscript reports the characterization of a little studied lipid binding protein of the perilipin family, perilipin 4 (Plin4). These proteins are characterized by 11-mer repeats that form amphipathic helices. All bind to lipid droplets but are differentially expressed in different tissues. Plin4 is primarily expressed in adipocytes and is distinguished by the very high number of 3X11-mer repeats. The physical basis for selective interaction with droplets of primarily neutral lipids and possible role in regulating lipid homeostasis and metabolism is not well understood. The physical characterization presented in this manuscript is a potentially important step in this understanding.

The data presented is very extensive with observations on the effect of length, hydrophobicity and charge on distributions in organelles of various cell types using numerous protein constructs. Aside from some CD-based characterization of helix formation in the presence of lipids, observations are primarily based on fluorescence observations on distribution to various organelles in different cell types. While the use of large fluorescent tags like GFP and mCherry that can be included in expression constructs is always susceptible to effects from the tags

themselves, the results are fairly reported and interpreted. The general conclusions regarding the interplay of mild hydrophobicity over a long helical segments and specific charge interactions leading to some selectivity for large neutral lipid droplets as opposed to lipid bilayer structures is likely valid. However, attention to some minor issues may improve the manuscript.

1) The presentation is very lengthy with significant repetition between results and discussion.

The discussion could be shortened. Also, there are a few instances where results are mentioned that seem to give minimal support to the general aims of the paper. For example, line 179 mentions fast recovery from photo bleaching is mentioned and reported in supplementary material with little additional discussion of its relevance. Instances like this could be removed.

2) A number of different cell-types are used with little justification of choice. Some justification could be given.

3) The structural characterization of the protein as a long amphipathic helix is in some cases rather qualitative. For example, when bound to oil droplets characterization is by resistance to trypsin cleavage. While it is clear that the system is difficult to characterize there may be ways to strengthen the argument.

4) It is also difficult to characterize protein concentration because of the lack of aromatic residues in the sequence. This is mentioned in the methods section and the “annex”, but it would be appropriate to put a cautionary note in the main text section where lipid to protein ratios are of importance.

Reviewer #3 (Remarks to the Author):

This manuscript by Copic and colleagues investigates how Plin4 is targeted to the surface of lipid droplets (LDs). The authors show that an unusual amphipathic helix consisting of 29 repeats of a 33aa unit is responsible for Plin4 LD targeting. The efficiency of Plin4 LD binding increases with the number of repeats suggesting cooperative mode of binding. Using a Plin4 construct with an intermediate AH length and poor binding to membranes (both bilayers and monolayers), the contribution of several parameters (such as charge, hydrophobicity, lipid packing, etc) was evaluated. These experiments showed that Plin4 AH has remarkable low affinity for membrane binding. Finally, it is shown that Plin4 LD localization appears to be a consequence of its strong interaction with neutral lipids and that it has the capacity to stabilize oil droplets both in vitro and in vivo. Understanding the mechanisms by which specific proteins are targeted to LDs while neglecting other cellular membranes is an important open question in cell biology. While focusing on an exceptional case, this study does make an important contribution to the field and that should be of interest to a general audience. However, there are a few points that should be addressed to strengthen some of the conclusions.

The first point concerns the levels of expression of the different constructs. The evaluation of

Plin4 AH in Figs 1, 3 and 4 concentrates on a variety of parameters however protein abundance is never considered (Fig 3d gives only a very rough estimate). Also, for some of the constructs the heterogeneity between cells (e. g. Fig 3e) appears dramatic. I wonder whether some of the effects observed may be due to differences in expression of the constructs analyzed. Moreover, in Fig. 4C, the % of LDs with wt protein is different on the 2 graphs (~25 vs 50%). It is unclear why under seemingly similar conditions the same protein appears to have so different LD localization efficiencies.

The second point concerns the dynamics of Plin4 association with LDs and the cooperativity between the repeats. Constructs with intermediate repeat lengths (12-20 mer) appear to have a very rapid turnover time in LDs induced by oleate-loading. This is based on the increased LD localization upon formaldehyde fixation and a FRAP experiment (Fig S2h). I would like to know whether the FRAP recovery time is also proportional to the number of repeats, as it should be according to the “Velcro” idea. Also, I wonder whether the mode of Plin4 interaction with LDs generated by CCT1 RNAi is similar to regular LDs. The result shown in Fig 7c,d is quite spectacular. However, it is not intuitive that such a degree of LD stabilization can be achieved by a highly dynamic protein coat. Thus, a more thorough analysis of the different length Plin4 on these LDs would be very informative.

Minor points:

- The yeast protein Pet10 was recently shown to be a perilipin-like protein. Sentence 150-151 should be revised.
- Why in Fig. 6d different repeat number Plin4 were mixed? Does longer repeats Plin4 versions outcompete shorter ones? It is mentioned that overexpression of different repeat length does not change LD size in vivo. What about in vitro?
- Line 408: delete the t in “tDrosophila Schneider”

Point-by-point response to reviewers' comments

General remarks.

Overall, we thank the reviewers for their careful analysis of our work and for their supportive comments. Below is a list of the general changes that we made. The point-by-point response then follows.

- We have added some additional references to the Introduction and we have made some small edits.
- We have adjusted the resolution of Suppl Fig. 1
- We have removed the FRAP data (panels g and h) from Suppl Fig. 2, which now ends with panel f.
- We have added an additional panel to Figure 5 (the new panel 5f) to include experiments with dioleoylglycerol as requested. The Figure legend has been adjusted accordingly.
- We have replaced Fig. 6e (previously 6d) with a figure using new data. The result is the same as before, but we now use the same fluorescent and non-fluorescent protein in the experiment (Plin4-12mer). We also added a new panel 6d to include data on the variability in our DLS measurements, which is important in connection with the new panel b in Suppl Fig. 5.
- We have added new data to Suppl Fig. 5:
 - panels a and b show a new experiment that demonstrates that vortexing does not change the properties of the Plin4 AH
 - panel c shows an EM image that was previously in this figure, except that the size has been reduced.
 - panel d shows a new experiment that demonstrates that the size of oil droplets formed by Plin4 AH, determined by DLS, does not correlate with the length of the AH.
- We show full gels and Western blot membranes corresponding to the gels shown in Fig. 6e, g, h and blots shown in Fig. 7b. These data can be found in the new Suppl Fig. 7.
- We have revised the Results section and the Figure legends to accommodate the changes listed above. We have also made some small deletions to shorten the text and some small edits.
- We have shortened the Discussion (to avoid repetition with the Results section, and following reviewers' comments, and we have removed the subheadings). We have added some additional references, and one reference was removed.
- in order to keep the ms within the size limits of the article format, we did not further extend the discussion (as suggested by reviewer 1) but rather shortened it (as suggested by reviewer 2). We also removed the FRAP data (as suggested by reviewer 2) instead of further studying Plin4 dynamics (as suggested by reviewer 3), which turns out to be complex (see our point-by-point response).
- The changes in the text are highlighted in the word document (except the deletions). We were using track changes to follow all small changes and to show the text that has been deleted, but we find that it is not possible to

submit a document with track changes through the submission system.
We are happy to send this document directly to the editor.

Reviewer #1 (Remarks to the Author):

The work by Copic et al. describes work to delineate the mechanism by which LD binding proteins selective bind to the LD surface. As model protein to use perilipin 4, which, when compared to all other perilipins, is highly unusual in its domain structure consisting predominantly of a repeating stretch of aa that may form an amphipathic alpha helix. The exact structure of this protein is currently unknown, as are its interactions with lipids. This work significantly contributes to our understanding of this important unresolved area of LD biology. That being said I do have some significant comments and suggestions for edits.

We thank the reviewer for careful reading of our manuscript, for the detailed comments and suggestions, and for the overall positive comments.

1. For of all the authors use the word “outstanding” to describe the alpha helix of perilipin 4 (in the abstract and throughout their paper). I do not like this terminology as it can easily be misinterpreted. The authors should change this word to either “unusual” or “exceptional in length and xxx”. Again, just the word outstanding in English has different meanings and may lead to misinterpretation of the authors meaning. Exceptional is better but then clarify in the text exactly in what way it is exceptional. Unusual is perhaps the least controversial replacement.

We have replaced the word “outstanding” with “exceptional” throughout the manuscript, as the reviewer suggests.

2. Line 63. This sentence at the end of the second paragraph on page 3 is incomplete.

It seems to us that the sentence is complete.

3. In the bottom paragraph of page 3 the authors refer to work done by the authors themselves and others on the interaction of AH domains with bilayer membranes. However, there has been extensive work performed by others on the interaction of AH domains with lipid monolayers. Specifically work on apolipoproteins but also, more recently, on perilipins. The authors need to cite this work and comment on how this work relates to their current study. Specifically the following two references are currently lacking in this work and need to be recognized:

Sletten A, Seline A, Rudd A, Logsdon M, Listenberger LL. Surface features of the lipid droplet mediate perilipin 2 localization. *Biochem Biophys Res Commun.* 2014 Sep 26;452(3):422-7. doi: 10.1016/j.bbrc.2014.08.097.

and

Mirheydari M, Rathnayake SS, Frederick H, Arhar T, Mann EK, Cocklin S, Kooijman EE. Insertion of perilipin 3 into a glycerophospholipid monolayer depends on lipid headgroup and acyl chain species. *J Lipid Res.* 2016 Aug;57(8):1465-76. doi: 10.1194/jlr.M068205.

Specifically this latter work discusses some aspects of perilipin interaction with LDs that need to be discussed in the current manuscript. The comment in this paragraph "It is not known which parameters are important for AH binding to the LD surface." is thus not entirely correct.

We have changed the wording of this sentence, and we have added the Mirheydari reference. We added the Sletten reference in the Discussion. These two papers address localization of perilipin 2 and perilipin 3 to LDs. Please note, however, that the Sletten study uses non purified Plin2 and that its binding to liposomes is assessed in a relative manner: the authors simply show that binding to DOPC liposomes is higher than on POPC liposomes, but the absolute binding is not determined and the investigation is limited to these two lipid compositions. Our study is much more complete: many lipid surfaces are tested and Plin4 binding properties are determined in an absolute manner. We do observe significantly better binding of Plin4 to DOPC vs POPC liposomes, but in both cases binding remains marginal as compared to what is observed with diphytanoyl liposomes.

Additionally there are other works showing how ER resident proteins end up on LDs. E.g. caveolin, and this work is also relevant for this paper.

We have focused in this paper on the targeting of amphipathic helices to LDs. We agree that our results may also have implications for the targeting of other proteins, for example proteins containing hairpin loops, but this will have to be tested in future work.

In short the discussion, while good, needs some expansion to incorporate what is known.

Results section:

4. The authors nicely discuss and show the aa sequence and predicted 3-11 structure of perilipin 4 and compare this to other perilipins. The figure in the supplementary results is hard to read though. Please enhance the resolution. Currently it is not acceptable for publication.

Done.

5. The authors use AH-GFP fusion proteins for their cellular work to determine LD binding. While the data seem convincing, I wonder if the fusion proteins themselves have a strong effect on LD / membrane binding? In other words the authors do not show if their constructs are soluble by themselves or if they form intracellular aggregates when not fused to GFP or another fluorescent protein. I thus miss a convincing control.

Fluorescent protein fusions are a standard and widely-used technique in cell

biology. Indeed, a fluorescent tag can to varying extents influence the localization/function of the tagged protein. We cannot test the localization of our untagged AHs in cells because there are no antibodies available. However, we disagree with the assessment of the reviewer that we have performed no controls of our fluorescent fusions.

i) We show that GFP/mCherry fused to related AH or non-AH sequences do not localize to LDs or any other visible structures in the cell (see Figures 1,3,4,7, S2). In addition, the GFP/mCherry alone appears completely soluble under all experimental conditions that we have used.

ii) We have constructed a large number of wild-type and mutant AHs. For each AH property tested (length, hydrophobicity, charge), we have prepared more than one construct (i.e., 5 for length, 5 for hydrophobicity – counting only 1 cell type!) and all results are consistent with each other.

iii) We have in fact expressed UNTAGGED AHs in bacterial cells and purified them to high purity. First, these experiments demonstrate that these AHs are soluble in bacterial cells (see Figure S3). If untagged AHs are soluble in bacterial cells, it does not seem likely that they would be forming aggregates in other cells unless fused to a fluorescent protein. Second, we perform a liposome binding experiment with untagged AHs of increasing length, obtaining a similar result as with tagged AHs in cells (compare Fig. 1e, f and Fig. 5f).

iv) We can also state that C-terminally tagged Plin4-4mer behaves quantitatively the same as N-terminally tagged Plin4-4mer in HeLa cells (our unpublished data).

Thus, we think that we can conclude with a fair certainty that the effects that we are seeing are largely reflecting the behavior of our AHs, not the tags.

5. On line 150 the authors suggest there are no yeast perilipins. This is in fact incorrect. Here is a paper the authors may want to consult: Gao Q, Binns DD, Kinch LN, Grishin NV, Ortiz N, Chen X, Goodman JM. Pet10p is a yeast perilipin that stabilizes lipid droplets and promotes their assembly. *J Cell Biol.* 2017 Aug 11. pii: jcb.201610013. doi: 10.1083/jcb.201610013

This paper appeared online after we had submitted our manuscript to Nature Communications (on July 14, 2017). We have now included this reference in our revised manuscript and have changed the text to read:

“We confirmed the lipid surface targeting of the Plin4 AH by expressing it in budding yeast, which contain proteins that are distantly related to human perilipins^{27,40}”.

We also cite this paper in some other places in our revised manuscript.

6. On page 9 the authors address the issue of charge for protein targeting. This issue appears to be highly complex and authors make some interesting points concerning protein and lipid charge in this respect. While I agree with the authors on most aspects of this point (the fact that its complex) I do feel the authors are missing part of the story. First is the observation that oil droplets by themselves are negatively charged in water (or any aqueous solution, see for example:

Ghimire C, Koirala D, Mathis MB, Kooijman EE, Mao H. Controlled particle

collision leads to direct observation of docking and fusion of lipid droplets in an optical trap. *Langmuir*. 2014 Feb 11;30(5):1370-5. doi: 10.1021/la404497v.

and references herein). Additionally, the acidic amino acid residues do not necessarily remain negatively charged when interacting with a lipid surface (or any surface). In fact the pKa of these residues depends sensitively on local conditions. While this may not be of effect here, I think it is important the authors consider these possibilities.

We agree with the reviewer on the importance of quoting the other suggested references (point 3), but we think that this *Langmuir* paper, while interesting, is too far from our topics. This paper studies the fusion of artificial oil droplets, either made of pure triolein or covered with PC (hence neutral or zwitterionic). The charge is carried by ions from the Hofmeister series to control water organization during droplet fusion. This process is quite far from what is studied here.

Overall, we agree with the reviewer that the question of the influence of charge on LD targeting is interesting and that there may be more subtle effects at play that we do not address here. However, we would like to point out that we have already performed a rather extensive mutagenesis, where we modulate the charge in three different ways; this has never been done before. The 2D to N mutation (which means in total mutating 8D to N in the full 4-mer repeat) has a strong effect on targeting even though the mutation is very mild and should largely affect charge but not other features (for example, helix propensity; see for example Pace & Scholz, 1998).

7. Top of page 10. Here the authors further discuss their results on their experiments with charged residues of perilipin 4. It would be nice to see a further discussion (here or in the discussion) on the relevant physics of lipid interfaces which is important but which the authors skip over. See my previous comment as well.

We agree with the reviewer that these are very interesting questions, but as this paper is already very long, we can hardly speculate more. Instead, we plan to revisit these questions in our future work, and we are very grateful to the reviewer for this encouragement.

8. The authors (on page 10 in section of liposome interactions) discuss work with lipids of different saturation. Others have previously shown that this is indeed important for perilipin interaction with lipid interfaces, but not in the way the authors postulate here (unsaturation perturbs perilipin 3 insertion, not increase it). See the reference (Mirheydari et al) given above for further details and additional references. The authors need to cite these works in their manuscript and comment on the results in relationship to these previously published works.

As explained above, we now quote the Mirheydari et al paper both in the Introduction and in the Discussion. However, throughout the text of this paper

and even in the conclusion, the authors acknowledge contradictions between their monolayer measurements and cellular observations that were previously published, both regarding the relative roles of the N terminal part vs C terminal part of the protein and the role of lipid unsaturation (they did observe a positive effect of lipid saturation). Perhaps, as acknowledged by the authors, the monolayer experiments do not properly mimic an LD interface. We do not refer to the Mirhedyari paper in the Results section, because our measurements with lipid bilayers or oil droplets cannot be directly compared with the measurements of Mirhedyari et al.

9. I do not understand why the authors have not used diacylglycerol in their liposome experiments. While DAGs can cause issues with liposome formation, small amounts can easily be incorporated when care in liposome preparation are taken. The authors have made a big deal about the curvature defects generated by DAG in the past so I was highly surprised to see work on a non-natural lipid in this regard. While the results with diphytanoyl lipids are certainly interesting they are perhaps not physiologically relevant. The authors need to include data on diacylglycerol, a lipid likely present in the lipid monolayer surrounding LDs, or at least justify in the paper why they did not use DAG. There is experimental evidence (from Nate Wollins group) that shows that DAG is the binding partner for perilipin 3.

We thank the reviewer for this suggestion. We now include an experiment with liposomes containing an increasing amount of DAG (see revised Figure 5). Strikingly, DAG also shows only a minor effect on Plin4 binding to liposomes, further highlighting the unique character of this AH.

We quote the results of the Wolins group (Skinner et al., JBC 2009), which suggest that DAG promotes binding of endogenous Plin3, and possibly also endogenous Plin4 to the ER surface in OP9 preadipocyte cells (the ER staining of Plin4 is much less convincing), at the resolution of conventional fluorescent microscopy and with the caveats of immunofluorescent staining. However, we would like to also point out a recent study (Ben M'Barek et al., Dev Cell 2017), which shows that DAG promotes formation of larger LDs that remain connected with the ER. Therefore, it is possible that what the Wolins group has observed is in fact coating of ER-associated LDs, not the ER bilayer. The Wolins group did not test binding of purified Plin3, Plin4 or Plin5 to DAG-containing liposomes in vitro, but only assayed binding of perilipins to LDs in cells or to rather crude subcellular fractions. Thus, there is no contradiction between our observation that DAG does not promote binding of the Plin4 AH to bilayers and what is reported in the literature. We have added references to these two studies to the revised Results section.

10. I am worried about the work with pure triolein that the authors present starting on page 12. While they do some controls and mention protein denaturation, they do NOT convincingly show that perilipin 4 is not denatured by their vortexing procedure. There are far better experiments that one can do that are convincing and preclude the possibility of protein denaturation. This section should be removed from the manuscript. So, the authors make a case for, but do not provide convincing proof, that denaturation is not a problem in these experiments.

We thank the reviewer for raising this concern. We now include an additional control experiment to convincingly show that the properties of the Plin4 AH are not affected by vortexing: the protein binds to diphytanoyl liposomes just as efficiently after as before vigorous vortexing (Supplementary figure 5a,b).

Discussion:

11. First paragraph. The authors need to be more careful when they interpret their results obtained from their oil experiments. These results are too preliminary and should be removed. The remainder of the paper is sufficient to be published without this.

We appreciate the suggestion that a large part of our data could be removed from this manuscript, however we don't see how this is possible. In Figure 5, we show that the Plin4 AH does not significantly interact with liposomes of different compositions, except for liposomes containing unusual branched (diphytanoyl) phospholipids. Given that these phospholipids are not present in eukaryotic cells, how can we then explain why this AH efficiently targets LDs in diverse cells? Our experiments with oil (Figure 6) give a clue, and we provide further support for these experiments with experiments in *Drosophila* S2 cells (Figure 7), which show that the Plin4 AH can replace phospholipids in cells, not just in vitro. We therefore think that the oil experiment is an integral part of this paper. Note also that the new control experiment with vortexing, as described in our response to Point 10 above, directly addresses the reviewer's concerns. Finally, we note that the other reviewers do not contest the merits of the oil experiment.

12. line 453. The authors need to cite the works mentioned above. And discuss their results in light of these previous works.

We have added additional references to this sentence.

13. Lines 484-486. This statement is preliminary and needs to be evaluated in light of the biophysics of the LD lipid interface. Some important aspects of which the authors do not mention in this paper. Which, of course, is ok as this is not their main focus. BUT they need to write this particular statement more carefully in light of other results and what is known of lipid interface electrostatics.

We have modified the Discussion to not over-state our observations regarding the influence of charge on LD targeting and to avoid repetition with the Results section. The paragraph containing lines 484-486 has been removed.

14. paragraph 2 on page 17. This should be removed or extensively rewritten. See my comments above.

We have rewritten this paragraph. Again, we want to point out that our new vortex experiment further validates our oil assay.

15. paragraph 3 on page 17. There are numerous papers from a Mexican group on the interaction of apolipoproteins and their peptides with lipid interfaces. This work may be relevant for the types of interactions the authors suggest here. The main authors of these papers are Xicohtencatl-Cortes, Mas-Oliva and Castillo.

There is indeed a vast amount of literature describing the interaction of apolipoproteins with different types of lipid interfaces, demonstrating that such complex and important questions can hardly be resolved in one single paper. We have studied carefully the work of above-mentioned and other authors. Due to space and length limitations, in particular the restricted number of references allowed, we are quoting the other papers suggested by this reviewer, but not the ones listed above.

Overall this is exciting new work and deserving of publication after significant revision.

We thank the reviewer again for the encouragement and for the detailed and insightful comments.

Reviewer #2 (Remarks to the Author):

This manuscript reports the characterization of a little studied lipid binding protein of the perilipin family, perilipin 4 (Plin4). These proteins are characterized by 11-mer repeats that form amphipathic helices. All bind to lipid droplets but are differentially expressed in different tissues. Plin4 is primarily expressed in adipocytes and is distinguished by the very high number of 3X11-mer repeats. The physical basis for selective interaction with droplets of primarily neutral lipids and possible role in regulating lipid homeostasis and metabolism is not well understood. The physical characterization presented in this manuscript is a potentially important step in this understanding.

The data presented is very extensive with observations on the effect of length, hydrophobicity and charge on distributions in organelles of various cell types using numerous protein constructs. Aside from some CD-based characterization of helix formation in the presence of lipids, observations are primarily based on fluorescence observations on distribution to various organelles in different cell types. While the use of large fluorescent tags like GFP and mCherry that can be included in expression constructs is always susceptible to effects from the tags themselves, the results are fairly reported and interpreted. The general conclusions regarding the interplay of mild hydrophobicity over a long helical segments and specific charge interactions leading to some selectivity for large neutral lipid droplets as opposed to lipid bilayer structures is likely valid. However, attention to some minor issues may improve the manuscript.

1) The presentation is very lengthy with significant repetition between results and discussion. The discussion could be shortened. Also, there are a few instances where results are mentioned that seem to give minimal support to the general aims of the paper. For example, line 179 mentions fast recovery

form photo bleaching is mentioned and reported in supplementary material with little additional discussion of its relevance. Instances like this could be removed.

We have removed the FRAP data and references to this data. We have shortened the Discussion as the reviewer suggests.

2) A number of different cell-types are used with little justification of choice. Some justification could be given.

The different cell-types used are justified in the following ways:

- HeLa cells (bottom of pg. 5): “We expressed different fragments of the protein as fluorescent protein fusions in HeLa cells, which do not express endogenous Plin4 (Fig. 1d,e){Hein:2015eh}.”
- Yeast (pg. 6, 2nd paragraph): “We confirmed the lipid surface targeting of the Plin4 AH by expressing it in budding yeast, which contain proteins that are distantly related to human perilipins. ... The fact that the same sequence is targeted to LDs in such evolutionarily distant organisms speaks to the universal nature of AH-LD surface interactions”
- S2 cells (pg. 14): “To test this hypothesis, we turned to *Drosophila* Schneider 2 (S2) cells, which have been used extensively to study factors that influence LD homeostasis{Guo:2008ij, Kraemer:2011cm, Wilfling:2014dh}, and where LD production can be strongly induced by exogenous addition of fatty acids (Fig. 7a).”

3) The structural characterization of the protein as a long amphipathic helix is in some cases rather qualitative. For example, when bound to oil droplets characterization is by resistance to trypsin cleavage. While it clear that the system is difficult to characterize there may be ways to strengthen the argument.

We have included an additional control experiment to show that vortexing does not affect the behavior of this amphipathic helix (Supplementary Figure S5).

We agree with the reviewer that the trypsin resistance experiment does not permit any conclusions regarding the type of structure that the protein adopts when in contact with oil; the experiment does demonstrate that the protein is more structured than in solution. We have been very careful to not overstate any conclusion in our text. Further structural characterization of this system is beyond the scope of this work, but is indeed a very interesting direction for future studies.

4) It is also difficult to characterize protein concentration because of the lack of aromatic residues in the sequence. This is mentioned in the methods section and the “annex”, but it would be appropriate to put a cautionary note in the main text section where lipid to protein ratios are of importance.

We have added a cautionary note to the Results section.

Reviewer #3 (Remarks to the Author):

This manuscript by Copic and colleagues investigates how Plin4 is targeted to the surface of lipid droplets (LDs). The authors show that an unusual amphipathic helix consisting of 29 repeats of a 33aa unit is responsible for Plin4 LD targeting. The efficiency of Plin4 LD binding increases with the number of repeats suggesting cooperative mode of binding. Using a Plin4 construct with an intermediate AH length and poor binding to membranes (both bilayers and monolayers), the contribution of several parameters (such as charge, hydrophobicity, lipid packing, etc) was evaluated. These experiments showed that Plin4 AH has remarkable low affinity for membrane binding. Finally, it is shown that Plin4 LD localization appears to be a consequence of its strong interaction with neutral lipids and that it has the capacity to stabilize oil droplets both in vitro and in vivo. Understanding the mechanisms by which specific proteins are targeted to LDs while neglecting other cellular membranes is an important open question in cell biology. While focusing on an exceptional case, this study does make an important contribution to the field and that should be of interest to a general audience. However, there are a few points that should be addressed to strengthen some of the conclusions.

The first point concerns the levels of expression of the different constructs. The evaluation of Plin4 AH in Figs 1, 3 and 4 concentrates on a variety of parameters however protein abundance is never considered (Fig 3d gives only a very rough estimate). Also, for some of the constructs the heterogeneity between cells (e. g. Fig 3e) appears dramatic. I wonder whether some of the effects observed may be due to differences in expression of the constructs analyzed. Moreover, in Fig. 4C, the % of LDs with wt protein is different on the 2 graphs (~25 vs50%). It is unclear why under seemingly similar conditions the same protein appears to have so different LD localization efficiencies.

There is indeed a large variability in protein expression levels between individual cells, as is generally the case with transient expressions in cell lines. These differences to some extent affect our quantifications and are probably the reason for the two different mean values for the same construct in Fig. 4c, as is explained below. However, we would like to first point out that differences between cells in the level of protein targeting to LDs are in no way dramatic: in Fig. 3e, it can be clearly observed that all LDs in all cells contain the fluorescent protein, regardless of the large differences in protein expression levels. This was the case for all of our constructs for which we report 100% of LD localization and this is completely reproducible between experiments, regardless of microscope settings and of experimenter performing the experiment (3 different people have performed these experiments over the course of a few years). Likewise, the constructs that display 0% LD localization have never been observed on LDs in any of the cells, regardless of protein expression levels and experimental conditions. Variability in the % of LDs with fluorescent protein is therefore only associated with the constructs that display an intermediate level of LD targeting.

We have taken great care to control for the variability in protein expression levels between cells: we used the same microscope settings to compare different constructs, which allowed us to analyze cells with an intermediate expression level. The mutants presented on the same graph were compared in the same experiments, and wt control was included in all experiments. However, because this study was performed over the course of a few years, the microscope settings have changed (as well as the experimenter performing the experiment). We think that these two factors are responsible for the different mean values obtained with the wild-type construct. The 2N and csw mutants were the last mutants that we analyzed, and the graph on the right in Fig. 4C was obtained 2 years after the graph on the left.

Whether a cell contains 25% or 50% of AH-positive LDs in our assays is informative only to the extent that it allows us to rank the different constructs in order of their LD affinity. These rankings are robust and consistent with the properties of the different constructs. In figure 4C, for example, we can say with high certainty that the order of affinity of different AHs for LDs is the following:

2V / 2N,2V / csw,2V > 2Q,2V > wt > 2N > 2Q / csw

We have considered normalizing our measurements to the value obtained with the wild-type construct in a given experiment (all our experiments contained the wild-type control), but this is not possible due to some constructs displaying 100% or 0% of LD targeting. We therefore feel that the way we represent our data is the most fair and informative.

The second point concerns the dynamics of Plin4 association with LDs and the cooperativity between the repeats. Constructs with intermediate repeat lengths (12-20 mer) appear to have a very rapid turnover time in LDs induced by oleate-loading. This is based on the increased LD localization upon formaldehyde fixation and a FRAP experiment (Fig S2h). I would like to know whether the FRAP recovery time is also proportional to the number of repeats, as it should be according to the "Velcro" idea. Also, I wonder whether the mode of Plin4 interaction with LDs generated by CCT1 RNAi is similar to regular LDs. The result shown in Fig 7c,d is quite spectacular. However, it is not intuitive that such a degree of LD stabilization can be achieved by a highly dynamic protein coat. Thus, a more thorough analysis of the different length Plin4 on these LDs would be very informative.

These are very good suggestions, and we have invested a lot of effort into FRAP experiments. However, we have finally decided to not include any of these experiments in our revised manuscript, because they are complex and the manuscript is already very long (as noted by the reviewers). Instead, we will address these questions in our follow-up work. [REDACTED]

[REDACTED]

[REDACTED]

[REDACTED]

Minor points:

- The yeast protein Pet10 was recently shown to be a perilipin-like protein. Sentence 150-151 should be revised.

We have included this reference.

- Why in Fig. 6d different repeat number Plin4 were mixed? Does longer repeats Plin4 versions outcompete shorter ones? It is mentioned that overexpression of different repeat length does not change LD size in vivo. What about in vitro?

We have replaced Fig. 6d with a new figure, in which labelled Plin4 is of the same length as unlabelled (12-mer). The results of the experiment are the same as before.

We have also tested the effect of AH length on the size of oil droplets produced in vitro by our vortexing procedure. These results are reported in Supplementary figure S5. Within the limits of our experiment (reproducibility of the vortexing reaction, resolution of DLS), we do not see any correlation between droplet size and AH length, in agreement with our observations in cells.

- Line 408: delete the t in “tDrosophila Schneider”

Thank you for noticing this typo.

Reviewers' Comments:

Reviewer #1 (Remarks to the Author):

This is a very careful and detailed study on the interaction of the AH segment of perilipin 4, a highly unusual perilipin. The authors have carefully considered my concerns and addressed most of them. While I would have liked more discussion on certain aspects of this work I agree with the authors that this can be accomplished in follow up work.

Overall then this work addresses an important problem in cell biology, and brings us a step closer in understanding the enigmatic proteins of the perilipin family. As well as the question of how cytosolic proteins recognize and bind specifically to the LD surface.

Reviewer #2 (Remarks to the Author):

The authors have responded in a detailed and convincing manner to the questions and suggestions reviewers made on the initial version of this manuscript. While the actual revisions are minimal, they have added important data (for example that on diacylglycerol), have removed the very preliminary mention of FRAP data, have added some key references and have made the discussion more concise. This seems adequate given the length constraints of a Nature Communication. The manuscript now clearly documents an important step in understanding the selective interactions of the exceptionally long Plin4 amphipathic helix with lipid droplets.

Reviewer #3 (Remarks to the Author):

I am satisfied with the revisions made by the authors and I would recommend the publication of this very nice study.

As minor points, I would suggest the authors to state clearly in the text that the LD targeting phenotypes of the different constructs used do not depend on their expression levels.

Also, it would be appropriate to reference and briefly contrast and compare the results described in this manuscript for Plin4 and the ones recently reported for the LD targeting of CCT alpha (Prevost and Sharp et al. 2018)

Point-by-point response

Reviewer #3 (Remarks to the Author):

I am satisfied with the revisions made by the authors and I would recommend the publication of this very nice study.

As minor points, I would suggest the authors to state clearly in the text that the LD targeting phenotypes of the different constructs used do not depend on their expression levels.

We state clearly in the Results section that the LD targeting phenotypes of the different constructs used do not depend on their expression levels.

Also, it would be appropriate to reference and briefly contrast and compare the results described in this manuscript for Plin4 and the ones recently reported for the LD targeting of CCT alpha (Prevost and Sharp et al. 2018)

We have added the reference Prevost and Sharp et al. 2018, and we briefly describe this new study and compare the results with our work. For this, we have added two sentences to the first paragraph of the Discussion, and two sentences at the end of the Discussion.